# Distribution-Free Statistical Dispersion Control for Societal Applications

**Zhun Deng**
zhun.d@columbia.edu
Columbia University

**Thomas P. Zollo**
tpz2105@columbia.edu
Columbia University

**Jake C. Snell**
js2523@princeton.edu
Princeton University

**Toniann Pitassi**
toni@cs.columbia.edu
Columbia University

**Richard Zemel**
zemel@cs.columbia.edu
Columbia University

## Abstract

Explicit finite-sample statistical guarantees on model performance are an important ingredient in responsible machine learning. Previous work has focused mainly on bounding either the expected loss of a predictor or the probability that an individual prediction will incur a loss value in a specified range. However, for many high-stakes applications it is crucial to understand and control the *dispersion* of a loss distribution, or the extent to which different members of a population experience unequal effects of algorithmic decisions. We initiate the study of distribution-free control of statistical dispersion measures with societal implications and propose a simple yet flexible framework that allows us to handle a much richer class of statistical functionals beyond previous work. Our methods are verified through experiments in toxic comment detection, medical imaging, and film recommendation.

## 1   Introduction

Learning-based predictive algorithms are widely used in real-world systems and have significantly impacted our daily lives. However, many algorithms are deployed without sufficient testing or a thorough understanding of likely failure modes. This is especially worrisome in high-stakes application areas such as healthcare, finance, and autonomous transportation. In order to address this critical challenge and provide tools for rigorous system evaluation prior to deployment, there has been a rise in techniques offering explicit and finite-sample statistical guarantees that hold for any unknown data distribution and black-box algorithm, a paradigm known as distribution-free uncertainty quantification (DFUQ). In [1], a framework is proposed for selecting a model based on bounds on expected loss produced using validation data. Subsequent work [30] goes beyond expected loss to provide distribution-free control for a class of risk measures known as quantile-based risk measures (QBRMs) [8]. This includes (in addition to expected loss): median, value-at-risk (VaR), and conditional value-at-risk (CVaR) [23]. For example, such a framework can be used to get bounds on the 80th percentile loss or the average loss of the 10% worst cases.

While this is important progress towards the sort of robust system verification necessary to ensure the responsible use of machine learning algorithms, in some scenarios measuring the expected loss or value-at-risk is not enough. As models are increasingly deployed in areas with long-lasting societal consequences, we should also be concerned with the dispersion of error across the population, or the extent to which different members of a population experience unequal effects of decisions made based on a model's prediction. For example, a system for promoting content on a social platform may offer less appropriate recommendations for the long tail of niche users in service of a small set

of users with high and typical engagement, as shown in [19]. This may be undesirable from both a business and societal point of view, and thus it is crucial to rigorously validate such properties in an algorithm prior to deployment and understand how the outcomes *disperse*. To this end, we offer a novel study providing rigorous distribution-free guarantees for a broad class of functionals including key measures of statistical dispersion in society. We consider both differences in performance that arise between different demographic groups as well as disparities that can be identified even if one does not have reliable demographic data or chooses not to collect them due to privacy or security concerns. Well-studied risk measures that fit into our framework include the Gini coefficient [33] and other functions of the Lorenz curve as well as differences in group measures such as the median [5]. See Figure 4 for a further illustration of loss dispersion.

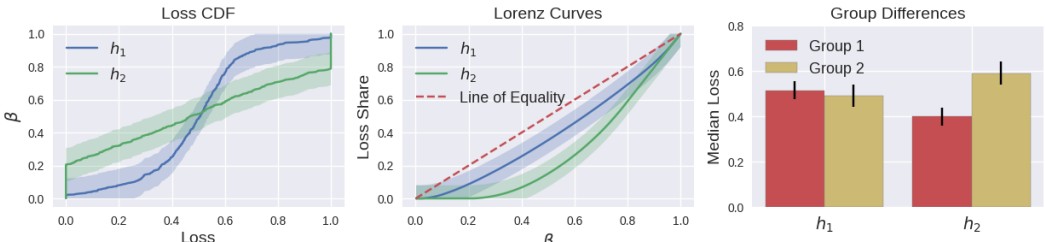

Figure 1: Example illustrating how two predictors (here $h_1$ and $h_2$) with the same expected loss can induce very different loss dispersion across the population. **Left**: The loss CDF produced by each predictor is bounded from below and above. **Middle**: The Lorenz curve is a popular graphical representation of inequality in some quantity across a population, in our case expressing the cumulative share of the loss experienced by the best-off $\beta$ proportion of the population. CDF upper and lower bounds can be used to bound the Lorenz curve (and thus Gini coefficient, a function of the shape of the Lorenz curve). Under $h_2$ the worst-off population members experience most of the loss. **Right**: Predictors with the same expected loss may induce different median loss for (possibly protected) subgroups in the data, and thus we may wish to bound these differences.

In order to provide rigorous guarantees for socially important measures that go beyond expected loss or other QBRMs, we provide two-sided bounds for quantiles and nonlinear functionals of quantiles. Our framework is simple yet flexible and widely applicable to a rich class of nonlinear functionals of quantiles, including Gini coefficient, Atkinson index, and group-based measures of inequality, among many others. Beyond our method for controlling this richer class of functionals, we propose a novel numerical optimization method that significantly tightens the bounds when data is scarce, extending earlier techniques [21, 30]. We conduct experiments on toxic comment moderation, detecting genetic mutations in cell images, and online content recommendation, to study the impact of our approach to model selection and tailored bounds.

To summarize our contributions, we: (1) initiate the study of distribution-free control of societal dispersion measures; (2) generalize the framework of [30] to provide bounds for nonlinear functionals of quantiles; (3) develop a novel optimization method that substantially tightens the bounds when data is scarce; (4) apply our framework to high-impact NLP, medical, and recommendation applications.

## 2    Problem setup

We consider a black-box model that produces an output $Z$ on every example. Our algorithm selects a predictor $h$, which maps an input $Z \in \mathcal{Z}$ to a prediction $h(Z) \in \hat{\mathcal{Y}}$. A loss function $\ell : \hat{\mathcal{Y}} \times \mathcal{Y} \to \mathbb{R}$ quantifies the quality of a prediction $\hat{Y} \in \hat{\mathcal{Y}}$ with respect to the target output $y \in \mathcal{Y}$. Let $(Z, Y)$ be drawn from an unknown joint distribution $\mathcal{P}$ over $\mathcal{Z} \times \mathcal{Y}$. We define the random variable $X^h := \ell(h(Z), Y)$ as the loss induced by $h$ on $\mathcal{P}$. The cumulative distribution function (CDF) of the random variable $X^h$ is $F^h(x) := \mathbb{P}(X^h \leq x)$. For brevity, we sometimes use $X$ and $F$ when we do not need to explicitly consider $h$. We define the inverse of a CDF (also called inverse CDF) $F$ as $F^-(p) = \inf\{x : F(x) \geq p\}$ for any $p \in \mathbb{R}$. Finally, we assume access to a set of validation samples $(Z, Y)_{1:n} = \{(Z_1, Y_1), \ldots, (Z_n, Y_n)\}$ for the purpose of achieving distribution-free CDF control with mild assumptions on the loss samples $X_{1:n}$. We emphasize that the "distribution-free"

requirement is on $(Z, Y)_{1:n}$ instead of $X_{1:n}$, because the loss studied on the validation dataset is known to us and we can take advantage of properties of the loss such as boundedness.

# 3 Statistical dispersion measures for societal applications

In this section, we motivate our method by studying some widely-used measures of societal statistical dispersion. There are key gaps between the existing techniques for bounding QBRMs and those needed to bound many important measures of statistical dispersion. We first define a QBRM:

**Definition 1** (Quantile-based Risk Measure). *Let $\psi(p)$ be a weighting function such that $\psi(p) \geq 0$ and $\int_0^1 \psi(p)\,dp = 1$. The quantile-based risk measure defined by $\psi$ is*

$$R_\psi(F) := \int_0^1 \psi(p) F^-(p) dp.$$

A QBRM is a linear functional of $F^-$, but quantifying many common group-based risk dispersion measures (e.g. Atkinson index) also involves forms like nonlinear functions of the (inverse) CDF or nonlinear functionals of the (inverse) CDF, and some (like maximum group differences) further involve nonlinear functions of functionals of the loss CDF. Thus a much richer framework for achieving bounds is needed here.

For clarity, we use $J$ as a generic term to denote either the CDF $F$ or its inverse $F^-$ depending on the context, and summarize the ***building blocks*** as below: **(i)** nonlinear functions of $J$, i.e. $\xi(J)$; **(ii)** functionals in the form of integral of nonlinear functions of $J$, i.e. $\int \psi(p)\xi(J(p))dp$ for a weight function $\psi$; **(iii)** composed functionals as nonlinear functions of functionals for the functional $T(J)$ with forms in **(ii)**, i.e. $\zeta(T(J))$ for a non-linear function $\zeta$.

## 3.1 Standard measures of dispersion

We start by introducing some classic non-group-based measures of dispersion. Those measures usually quantify wealth or consumption inequality *within* a social group (or a population) instead of quantifying differences among groups. Note that for all of these measures we only consider non-negative losses $X$, and assume that $\int_0^1 F^-(p)dp > 0$ [1].

**Gini family of measures.** Gini coefficient [33, 34] is a canonical measure of statistical dispersion, used for quantifying the uneven distribution of resources or losses. It summarizes the Lorenz curve introduced in Figure 4. From the definition of Lorenz curve, the greater its curvature is, the greater inequality there exists; the Gini coefficient is measuring the ratio of the area that lies between the line of equality (the $45°$ line) and the Lorenz curve to the total area under the line of equality.

**Definition 2** (Gini coefficient). *For a non-negative random variable $X$, the Gini coefficient is*

$$\mathcal{G}(X) := \frac{\mathbb{E}|X - X'|}{2\mathbb{E}X} = \frac{\int_0^1 (2p - 1)F^-(p)dp}{\int_0^1 F^-(p)dp},$$

*where $X'$ is an independent copy of $X$. $\mathcal{G}(X) \in [0, 1]$, with 0 indicating perfect equality.*

Because of the existence of the denominator in the Gini coefficient calculation, unlike in QBRM we need both an upper and a lower bound for $F^-$ (see Section 4.1.1). In the appendix, we also introduce the extended Gini family.

**Atkinson index.** The Atkinson index [2, 19] is another renowned dispersion measure defined on the non-negative random variable $X$ (e.g., income, loss), and improves over the Gini coefficient in that it is useful in determining which end of the distribution contributes most to the observed inequality by choosing an appropriate inequality-aversion parameter $\varepsilon \geq 0$. For instance, the Atkinson index becomes more sensitive to changes at the lower end of the income distribution as $\varepsilon$ increases.

**Definition 3** (Atkinson index). *For a non-negative random variable $X$, for any $\varepsilon \geq 0$, the Atkinson index is defined as the following if $\varepsilon \neq 1$:*

$$\mathcal{A}(\varepsilon, X) := 1 - \frac{(\mathbb{E}[X^{1-\varepsilon}])^{\frac{1}{1-\varepsilon}}}{\mathbb{E}[X]} = 1 - \frac{\left(\int_0^1 (F^-(p))^{1-\varepsilon} dp\right)^{\frac{1}{1-\varepsilon}}}{\int_0^1 F^-(p)dp}.$$

---

[1]This assumption is included for concise description, but not necessary. We also set $0/0 = 0$.

*And for $\varepsilon = 1$, $\mathcal{A}(1, X) := \lim_{\varepsilon \to 1} \mathcal{A}(\varepsilon, X)$, which will converge to a form involving the geometric mean of $X$. $\mathcal{A}(\varepsilon, X) \in [0, 1]$, and $0$ indicates perfect equality (see appendix for details).*

The form of Atkinson index includes a nonlinear function of $F^-$, i.e. $(F^-)^{1-\varepsilon}$, but this type of nonlinearity is easy to tackle since the function is monotonic w.r.t. the range of $F^-$ (see Section 4.2.1).

**Remark 1.** *The reason we study the CDF of $X$ and not $X^{1-\varepsilon}$ is that it allows us to simultaneously control the Atkinson index for all $\varepsilon$'s.*

In addition, there are many other important measures of dispersion involving more complicated types of nonlinearity such as the quantile of extreme observations and mean of range. Those measures are widely used in forecasting weather events or food supply. We discuss and formulate these dispersion measures in the appendix.

### 3.2 Group-based measures of dispersion

Another family of dispersion measures refer to minimizing differences in performance across possibly overlapping groups in the data defined by (protected) attributes like race and gender. Under equal opportunity [11], false positive rates are made commensurate, while equalized odds [11] aims to equalize false positive rates and false negative rates among groups. More general attempts to induce fairly-dispersed outcomes include CVaR fairness [32] and multi-calibration [14, 15]. Our framework offers the flexibility to control a wide range of measures of a group CDF $F_g$, i.e. $T(F_g)$, as well as the variation of $T(F_g)$ between groups. As an illustration of the importance of such bounds, [5] finds that the median white family in the United States has eight times as much wealth as the median black family; this motivates a dispersion measure based on the difference in group medians.

**Absolute/quadratic difference of risks and beyond.** The simplest way to measure the dispersion of a risk measure (like median) between two groups are quantities such as $|T(F_g) - T(F_{g'})|$ or $[T(F_g) - T(F_{g'})]^2$. Moreover, one can study $\xi(T(F_g) - T(F_{g'}))$ for some general nonlinear functions. These types of dispersion measures are widely used in algorithmic fairness [11, 20].

**CVaR-fairness risk measure and its extensions.** In [32], the authors further consider a distribution for each group, $\mathcal{P}_g$, and a distribution over group indices, $\mathcal{P}_{\text{Idx}}$. Letting $CVaR_{\alpha, \mathcal{P}_Z}(Z) := \mathbb{E}_{Z \sim \mathcal{P}_Z}[Z | Z > \alpha]$ for any distribution $\mathcal{P}_Z$, they define the following dispersion for the expected loss of group $g$ (i.e. $\mu_g := \mathbb{E}_{X \sim \mathcal{P}_g}[X]$) for $\alpha \in (0, 1)$:

$$\mathcal{D}_{CV, \alpha}(\mu_g) := CVaR_{\alpha, \mathcal{P}_{\text{Idx}}}\big(\mu_g - \mathbb{E}_{g \sim \mathcal{P}_{\text{Idx}}}[\mu_g]\big).$$

A natural extension would be $\mathcal{D}_{CV, \alpha}(T(F_g))$ for general functional $T(F_g)$, which we can write in a more explicit way [23]:

$$\mathcal{D}_{CV, \alpha}(T(F_g)) = \min_{\rho \in \mathbb{R}} \left\{ \rho + \frac{1}{1 - \alpha} \cdot \mathbb{E}_{g \sim \mathcal{P}_{\text{Idx}}}[T(F_g) - \rho]_+ \right\} - \mathbb{E}_{g \sim \mathcal{P}_{\text{Idx}}}[T(F_g)].$$

The function $[T(F_g) - \rho]_+$ is a nonlinear function of $T(F_g)$, but it is a monotonic function when $\rho$ is fixed and its further composition with the expectation operation is still monotonic, which can be easily dealt with.

**Uncertainty quantification of risk measures.** In [4], the authors study the problem of uncertainty of risk assessment, which has important consequences for societal measures. They formulate a deviation-based approach to quantify uncertainty for risks, which includes forms like: $\rho_\xi(\mathcal{P}_{\text{Idx}}) := \mathbb{E}_{g \sim \mathcal{P}_{\text{Idx}}}[\xi(T(F_g))]$ for different types of nonlinear functions $\xi$. Examples include variance uncertainty quantification, where $\mathbb{E}_{g \sim \mathcal{P}_{\text{Idx}}}\big(T(F_g) - \mathbb{E}_{g \sim \mathcal{P}_{\text{Idx}}}T(F_g)\big)^2$; and $\mathbb{E}_\psi[\xi(F^-(\alpha))] := \int_0^1 \xi(F^-(\alpha))\psi(\alpha)d\alpha$ to quantify how sensitive the $\alpha$-VaR value is w.r.t its parameter $\alpha$ for some non-negative weight function $\psi$.

## 4  Distribution-free control of societal dispersion measures

In this section, we introduce a simple yet general framework to obtain rigorous upper bounds on the statistical dispersion measures discussed in the previous section. We will provide a high-level summary of our framework in this section, and leave detailed derivations and most examples to the

appendix. Our discussion will focus on quantities related to the inverse of CDFs, but similar results could be obtained for CDFs.

In short, our framework involves two steps: produce upper and lower bounds on the CDF (and thus inverse CDF) of the loss distribution, and use these to calculate bounds on a chosen target risk measure. First, we will describe our extension of the one-sided bounds in [30] to the two-sided bounds necessary to control many societal dispersion measures of interest. Then we will describe how these CDF bounds can be post-processed to provide control on risk measures defined by nonlinear functions and functionals of the CDF. Finally, we will offer a novel optimization method for tightening the bounds for a chosen, possibly complex, risk measure.

## 4.1 Methods to obtain confidence two-sided bounds for CDFs

For loss values $\{X_i\}_{i=1}^n$, let $X_{(1)} \leq \ldots \leq X_{(n)}$ denote the corresponding order statistics. For the uniform distribution over [0,1], i.e. $\mathcal{U}(0, 1)$, let $U_1, \ldots, U_n \sim^{iid} \mathcal{U}(0, 1)$ denote the corresponding order statistics $U_{(1)} \leq \ldots \leq U_{(n)}$. We will also make use of the following:

**Proposition 1.** *For the CDF $F$ of $X$, if there exists two CDFs $F_U, F_L$ such that $F_U \succeq F \succeq F_L$[2], then we have $F_L^- \succeq F^- \succeq F_U^-$.*

We use $(\hat{F}_{n,L}^\delta, \hat{F}_{n,U}^\delta)$ to denote a $(1 - \delta)$-confidence bound pair $((1 - \delta)$-CBP), which satisfies $\mathbb{P}(\hat{F}_{n,U}^\delta \succeq F \succeq \hat{F}_{n,L}^\delta) \geq 1 - \delta$.

We extend the techniques developed in [30], wherein one-sided (lower) confidence bounds on the uniform order statistics are used to bound $F$. This is done by considering a one-sided minimum goodness-of-fit (GoF) statistic of the following form: $S := \min_{1 \leq i \leq n} s_i(U_{(i)})$, where $s_1, \ldots, s_n : [0, 1] \rightarrow \mathbb{R}$ are right continuous monotone nondecreasing functions. Thus, $\mathbb{P}(\forall i : F(X_{(i)}) \geq s_i^-(s_\delta)) \geq 1 - \delta$, for $s_\delta = \inf_r\{r : \mathbb{P}(S \geq r) \geq 1 - \delta\}$. Given this step function defined by $s_1, \ldots, s_n$, it is easy to construct $\hat{F}_{n,L}^\delta$ via conservative completion of the CDF. [30] found that a Berk-Jones bound could be used to choose appropriate $s_i$'s, and is typically much tighter than using the Dvoretzky–Kiefer–Wolfowitz (DKW) inequality to construct a bound.

### 4.1.1 A reduction approach to constructing upper bounds of CDFs

Now we show how we can leverage this approach to produce two-sided bounds. In the following lemma we show how a CDF upper bound can be reduced to constructing lower bounds.

**Lemma 1.** *For $0 \leq L_1 \leq L_2 \cdots \leq L_n \leq 1$, if $\mathbb{P}(\forall i : F(X_{(i)}) \geq L_i) \geq 1 - \delta$, then, we have*

$$\mathbb{P}(\forall i : \lim_{\epsilon \to 0^+} F(X_{(i)} - \epsilon) \leq 1 - L_{n-i+1}) \geq 1 - \delta.$$

*Furthermore, let $R(x)$ be defined as $1 - L_n$ if $x < X_{(1)}$; $1 - L_{n-i+1}$ if $X_{(i)} \leq x < X_{(i+1)}$ for $i \in \{1, 2, \cdots, n - 1\}$; $1$ if $X_{(n)} \leq x$. Then, $F \preceq R$.*

Thus, we can simultaneously obtain $(\hat{F}_{n,L}^\delta, \hat{F}_{n,U}^\delta)$ by setting $L_i = s_i^-(s_\delta)$ and applying (different) CDF conservative completions. In practice, the CDF upper bound can be produced via post-processing of the lower bound. One clear advantage of this approach is that it avoids the need to independently produce a pair of bounds where each bound must hold with probability $1 - \delta/2$.

## 4.2 Controlling statistical dispersion measures

Having described how to obtain the CDF upper and lower bounds $(\hat{F}_{n,L}^\delta, \hat{F}_{n,U}^\delta)$, we next turn to using these to control various important risk measures such as the Gini coefficient and group differences. We will only provide high-level descriptions here and leave details to the appendix.

### 4.2.1 Control of nonlinear functions of CDFs

First we consider bounding $\xi(F^-)$, which maps $F^-$ to another function of $\mathbb{R}$.

---

[2]We also loosely consider a non-negative increasing function $\tilde{F}$ such that $\lim_{x \to \infty} \tilde{F}(x) = 1$, but $\tilde{F}(x) < 1$ for any real value $x$, as a CDF.

**Control for a monotonic function.** We start with the simplest case, where $\xi$ is a monotonic function in the range of $X$. For example, if $\xi$ is an increasing function, and with probability at least $1 - \delta$, $\hat{F}^\delta_{n,U} \succeq F \succeq \hat{F}^\delta_{n,L}$; then further by Proposition 1, we have that $\xi(\hat{F}^{\delta,-}_{n,L}) \succeq \xi(\hat{F}^-) \succeq \xi(\hat{F}^{\delta,-}_{n,U})$ holds with probability at least $1 - \delta$. This property could be utilized to provide bounds for the Gini coefficient or Atkinson index by controlling the numerator and denominator separately as integrals of monotonic functions of $F^-$.

**Example 1** (Gini coefficient). *If given a $(1 - \delta)$-CBP $(\hat{F}^\delta_{n,L}, \hat{F}^\delta_{n,U})$ and $\hat{F}^\delta_{n,L} \succeq 0$ [3], we can provide the following bound for the Gini coefficient. Notice that*

$$\mathcal{G}(X) = \frac{\int_0^1 (2p - 1)F^-(p)dp}{\int_0^1 F^-(p)dp} = \frac{\int_0^1 2pF^-(p)dp}{\int_0^1 F^-(p)dp} - 1.$$

*Given $F^-(p) \geq 0$ (since we only consider non-negative losses, i.e. $X$ is always non-negative), we know*

$$\mathcal{G}(X) \leq \frac{\int_0^1 2p\hat{F}^{\delta,-}_{n,L}(p)dp}{\int_0^1 \hat{F}^{\delta,-}_{n,U}(p)dp} - 1,$$

*with probability at least $1 - \delta$.*

**Control for absolute and polynomial functions.** Many societal dispersion measures involve absolute-value functions, e.g., the Hoover index or maximum group differences. We must also control polynomial functions of inverse CDFs, such as in the CDFs of extreme observations. For any polynomial function $\phi(s) = \sum_{k=0} \alpha_k s^k$, if $k$ is odd, $s^k$ is monotonic w.r.t. $s$; if $k$ is even, $s^k = |s|^k$. Thus, we can group $\alpha_k s^k$ according to the sign of $\alpha_k$ and whether $k$ is even or odd, and flexibly use the upper and lower bounds already established for the absolute value function and monotonic functions to obtain an overall upper bound.

**Example 2.** *If we have $(T^\delta_L(F_g), T^\delta_U(F_g))$ such that $T^\delta_L(F_g) \leq T(F_g) \leq T^\delta_U(F_g)$ holds for all $g$ we consider, then we can provide high probability upper bounds for*

$$\xi(T(F_{g_1}) - T(F_{g_2}))$$

*for any polynomial functions or the absolute function $\xi$. For example, with probability at least $1 - \delta$*

$$|T(F_{g_1}) - T(F_{g_2})| \leq \max\{|T^\delta_U(F_{g_1}) - T^\delta_L(F_{g_2})|, |T^\delta_L(F_{g_1}) - T^\delta_U(F_{g_2})|\}.$$

**Control for a general function.** To handle general nonlinearities, we introduce the class of functions of bounded total variation. Roughly speaking, if a function is of bounded total variation on an interval, it means that its range is bounded on that interval. This is a very rich class including all continuously differentiable or Lipchitz continuous functions. The following theorem shows that such functions can always be decomposed into two monotonic functions.

**Theorem 1.** *For $(\hat{F}^\delta_{n,L}, \hat{F}^\delta_{n,U})$, if $\xi$ is a function with bounded total variation on the range of $X$, there exists increasing functions $f_1, f_2$ with explicit and calculable forms, such that with probability at least $1 - \delta$, $\xi(F^-) \preceq f_1(\hat{F}^{\delta,-}_{n,L}) - f_2(\hat{F}^{\delta,-}_{n,U})$.*

As an example, recall that [4] studies forms like

$$\int_0^1 \xi(F^-(\alpha))\psi(\alpha)d\alpha$$

to quantify how sensitive the $\alpha$-VaR value is w.r.t its parameter $\alpha$. For nonlinear functions beyond polynomials, consider the example where $\xi = e^x + e^{-x}$. This can be readily bounded since it is a mixture of monotone functions.

---

[3]This can be easily achieved by taking truncation over 0. Also, the construction of $\hat{F}^\delta_{n,L}$ in Section A.2 always satisies this constraint.

#### 4.2.2 Control of nonlinear functionals of CDFs and beyond

Finally, we show how our techniques from the previous section can be applied to handle the general form $\int_0^1 \xi(T(F^-(\alpha)))\psi(\alpha)d\alpha$, where $\psi$ can be a general function (not necessarily non-negative) and $\xi$ can be a *functional* of the inverse CDF. To control $\xi(T(F^-))$, we can first obtain two-sided bounds for $T(F^-)$ if $T(F^-)$ is in the class of QBRMs or in the form of $\int \psi(p)\xi_2(F^-(p))dp$ for some nonlinear function $\xi_2$ (as in [4]). We can also generalize the weight functions in QBRMs from non-negative to general weight functions once we notice that $\psi$ can be decomposed into two non-negative functions, i.e. $\psi = \max\{\psi, 0\} - \max\{-\psi, 0\}$. Then, we can provide upper bounds for terms like $\int \max\{\psi, 0\}\xi(F^-(p))dp$ by adopting an upper bound for $\xi(F^-)$.

### 4.3 Numerical optimization towards tighter bounds for statistical functionals

Having described our framework for obtaining CDF bounds and controlling rich families of risk measures, we return to the question of how to produce the CDF bounds. One drawback of directly using the bound returned by Berk-Jones is that it is not weight function aware, i.e., it does not leverage knowledge of the target risk measures. This motivates the following numerical optimization method, which shows significant improvement over previous bounds including DKW and Berk-Jones bounds (as well as the truncated version proposed in [30]).

Our key observation is that for any $0 \leq L_1 \leq \cdots \leq L_n \leq 1$, we have $\mathbb{P}\big(\forall i,\ F(X_{(i)}) \geq L_i\big) \geq n! \int_{L_n}^1 dx_n \int_{L_{n-1}}^{x_n} dx_{n-1} \cdots \int_{L_1}^{x_2} dx_1$, where the right-hand side integral is a function of $\{L_i\}_{i=1}^n$ and its partial derivatives can be calculated exactly by the package in [21]. Consider controlling $\int_0^1 \psi(p)F^-(p)dp$ as an example. For any $\{L_i\}_{i=1}^n$ satisfying $\mathbb{P}\big(\forall i,\ F(X_{(i)}) \geq L_i\big) \geq 1 - \delta$, one can use conservative CDF completion to obtain $\hat{F}_{n,L}^\delta$, i.e.

$$\int_0^1 \psi(p)\xi(\hat{F}_{n,L}^{\delta,-}(p))dp = \sum_{i=1}^{n+1} \xi(X_{(i)}) \int_{L_{i-1}}^{L_i} \psi(p)dp,$$

where $L_{n+1}$ is 1, $L_0 = 0$, and $X_{(n+1)} = \infty$ or a known upper bound for $X$. Then, we can formulate tightening the upper bound as an optimization problem:

$$\min_{\{L_i\}_{i=1}^n} \sum_{i=1}^{n+1} \xi(X_{(i)}) \int_{L_{i-1}}^{L_i} \psi(p)dp$$

such that $\mathbb{P}\big(\forall i,\ F(X_{(i)}) \geq L_i\big) \geq 1 - \delta$, and $0 \leq L_1 \leq \cdots \leq L_n \leq 1$. We optimize the above problem with gradient descent and a simple post-processing procedure to make sure the obtained $\{\hat{L}_i\}_{i=1}^n$ strictly satisfy the above constraints. In practice, we re-parameterize $\{L_i\}_{i=1}^n$ with a network $\phi_\theta$ that maps $n$ random seeds to a function of the $L_i$'s, and transform the optimization objective from $\{L_i\}_{i=1}^n$ to $\theta$. We find that a simple parameterized neural network model with 3 fully-connected hidden layers of dimension 64 is enough for good performance and robust to hyper-parameter settings.

$$\gamma^* = \inf\{\gamma : n!\upsilon(L_1(\hat{\theta}) - \gamma, \cdots, L_n(\hat{\theta}) - \gamma, 1) \geq 1 - \delta, \gamma \geq 0\}.$$

Notice that there is always a feasible solution. We can use binary search to efficiently find (a good approximate of) $\gamma^*$.

## 5 Experiments

With our experiments, we aim to examine the contributions of our methodology in two areas: bound formation and responsible model selection.

### 5.1 Learn then calibrate for detecting toxic comments

Using the CivilComments dataset [6], we study the application of our approach to toxic comment detection under group-based fairness measures. CivilComments is a large dataset of online comments

labeled for toxicity as well as the mention of protected sensitive attributes such as gender, race, and religion. Our loss function is the Brier Score, a proper scoring rule that measures the accuracy of probabilistic predictions, and we work in the common setting where a trained model is calibrated post-hoc to produce confidence estimates that are more faithful to ground-truth label probabilities. We use a pre-trained toxicity model and apply a Platt scaling model controlled by a single parameter to optimize confidence calibration. Our approach is then used to select from a set of hypotheses, determined by varying the scaling parameter in the range $[0.25, 2]$ (where scaling parameter 1 recovers the original model). See the Appendix for more details on the experimental settings and our bound optimization technique.

### 5.1.1 Bounding complex expressions of group dispersion

First, we investigate the full power of our framework by applying it to a complex statistical dispersion objective. Our overall loss objective considers both expected mean across groups as well as the maximum difference between group medians, and can be expressed as: $\mathcal{L} = \mathbb{E}_g[T_1(F_g)] + \lambda \sup_{g,g'} |T_2(F_g) - T_2(F_{g'})|$, where $T_1$ is expected loss and $T_2$ is a smoothed version of a median (centered around $\beta = 0.5$ with spread parameter $a = 0.1$). Groups are defined by intersectional attributes: $g \in G = \{$black female, white female, black male, white male$\}$. We use 100 and 200 samples from each group, and select among 50 predictors. For each group, we use our numerical optimization framework to optimize a bound on $\mathcal{O} = T_1(F_g) + T_2(F_g)$ using the predictor (and accompanying loss distribution) chosen under the Berk-Jones method. Results are shown in Table 1. We compare our numerically-optimized bound (NN-Opt.) to the bound given by Berk-Jones as well as an application of the DKW inequality to lower-bounding a CDF.

Our framework enables us to choose a predictor that fits our specified fairness criterion, and produces reasonably tight bounds given the small sample size and the convergence rate of $\frac{1}{\sqrt{n}}$. Moreover, there is a large gain in tightness from numerical optimization in the case where $n = 100$, especially with respect to the bound on the maximum difference in median losses (0.076 vs. 0.016). These results show that a single bound can be flexibly optimized to improve on multiple objectives at once via our numerical method, a key innovation point for optimizing bounds reflecting complex societal concerns like differences in group medians [5].

Table 1: Applying our full framework to control an objective considering expected group loss as well as a maximum difference in group medians for $n = 100$ and $n = 200$ samples.

| Method | $n = 100$ | | | $n = 200$ | | |
| | Exp. Grp. | Max Diff. | Total | Exp. Grp. | Max Diff. | Total |
| --- | --- | --- | --- | --- | --- | --- |
| DKW | 0.36795 | 0.90850 | 1.27645 | 0.32236 | 0.96956 | 1.29193 |
| BJ | 0.34532 | 0.07549 | 0.42081 | 0.31165 | 0.00666 | 0.31831 |
| NN-Opt. (ours) | **0.32669** | **0.01612** | **0.34281** | **0.30619** | **0.00292** | **0.30911** |
| Empirical | 0.20395 | 0.00004 | 0.20399 | 0.20148 | 0.00010 | 0.20158 |

### 5.1.2 Optimizing bounds on measures of group dispersion

Having studied the effects of applying the full framework, we further investigate whether our method for numerical optimization can be used to get tight and flexible bounds on functionals of interest. First, $\beta$-CVaR is a canonical tail measure, and we bound the loss for the worst-off $1 - \beta$ proportion of predictions (with $\beta = 0.75$). Next, we bound a specified interval of the VaR ($[0.5, 0.9]$), which is useful when a range of quantiles of interest is known but flexibility to answer different queries within the range is important. Finally, we consider a worst-quantile weighting function $\psi(p) = p$, which penalizes higher loss values on higher quantiles, and study a smooth delta function around $\beta = 0.5$, a more robust version of a median measure. We focus on producing bounds using only 100 samples from a particular intersectionally-defined protected group, in this case black females, and all measures are optimized with the same hyperparameters. The bounds produced via numerical optimization (NN-Opt.) are compared to the bounds in [30] (as DKW has been previously shown to produce weak CDF bounds), including the typical Berk-Jones bound as well as a truncated version tailored to particular quantile ranges. See Table 2 and the Appendix for results.

The numerical optimization method induces much tighter bounds than Berk-Jones on all measures, and also improves over the truncated Berk-Jones where it is applicable. Further, whereas the truncated Berk-Jones bound will give trivial control outside of $[\beta_{min}, \beta_{max}]$, the numerically-optimized bound not only retains a reasonable bound on the entire CDF, but even improves on Berk-Jones with respect to the bound on expected loss in all cases. For example, after adapting to CVaR, the numerically-optimized bound gives a bound on the expected loss of 0.23, versus 0.25 for Berk-Jones and 0.50 for Truncated Berk-Jones. Thus numerical optimization produces both the best bound in the range of interest as well as across the rest of the distribution, showing the value of adapting the bound to the particular functional and loss distribution while still retaining the distribution-free guarantee.

Table 2: Optimizing bounds on measures for protected groups.

| Method | CVaR | VaR-Interval | Quantile-Weighted | Smoothed-Median |
|---|---|---|---|---|
| Berk-Jones | 0.91166 | 0.38057 | 0.19152 | 0.00038 |
| Truncated Berk-Jones | 0.86379 | 0.34257 | - | - |
| NN-Opt. (ours) | **0.85549** | **0.32656** | **0.17922** | **0.00021** |

## 5.2 Investigating bounds on standard measures of dispersion

Next, we aim to explore the application of our approach to responsible model selection under non-group-based fairness measures, and show how using our framework leads to a more balanced distribution of loss across the population. Further details for both experiments can be found in the Appendix.

### 5.2.1 Controlling balanced accuracy in detection of genetic mutation

RxRx1 [31] is a task where the input is a 3-channel image of cells obtained by fluorescent microscopy, the label indicates which of 1,139 genetic treatments the cells received, and there is a batch effect that creates a challenging distribution shift across domains. Using a model trained on the train split of the RxRx1 dataset, we evaluate our method with an out-of-distribution validation set to highlight the distribution-free nature of the bounds. We apply a threshold to model output in order to produce prediction sets, or sets of candidate labels for a particular task instance. Prediction sets are scored with a balanced accuracy metric that equally weights sensitivity and specificity, and our overall objective is: $\mathcal{L} = T_1(F) + \lambda T_2(F)$, where $T_1$ is expected loss, $T_2$ is Gini coefficient, and $\lambda = 0.2$. We choose among 50 predictors (i.e. model plus threshold) and use 2500 population samples to produce our bounds. Results are shown in Figure 2.

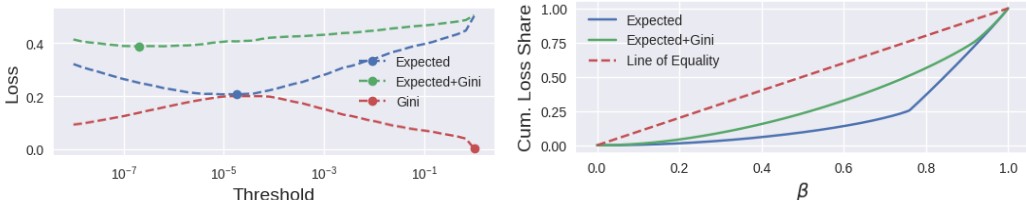

Figure 2: Left: Bounds on the expected loss, scaled Gini coefficient, and total objective across different hypotheses. Right: Lorenz curves induced by choosing a hypothesis based on the expected loss bound versus the bound on the total objective. The y-axis shows the cumulative share of the loss that is incurred by the best-off $\beta$ proportion of the population, where a perfectly fair predictor would produce a distribution along the line $y = x$.

The plot on the left shows how the bounds on the expected loss $T_1$, scaled Gini coefficient $\lambda T_2$, and total objective $\mathcal{L}$ vary across the different hypotheses (i.e. model and threshold combination for producing prediction sets). The bold points indicate the optimal threshold choice for each quantity. On the right is shown the Lorenz curves (a typical graphical expression of Gini) of the loss distributions induced by choosing a hypothesis based on the expected loss bound versus the bound on the total objective. Incorporating the bound on Gini coefficient in hypothesis selection leads to a more equal loss distribution. Taken together, these figures illustrate how the ability to bound a non-group based

dispersion measure like the Gini coefficient can lead to less skewed outcomes across a population, a key goal in societal applications.

### 5.2.2 Producing recommendation sets for the whole population

Using the MovieLens dataset [12], we test whether better control on another important non-group based dispersion measure, the Atkinson index (with $\epsilon = 0.5$), leads to a more even distribution of loss across the population. We train a user/item embedding model, and compute a loss that balances precision and recall for each set of user recommendations. Results are shown in Figure 3. Tighter control of the Atkinson index leads to a more dispersed distribution of loss across the population, even for subgroups defined by protected attributes like age and gender that are unidentified for privacy or security reasons.

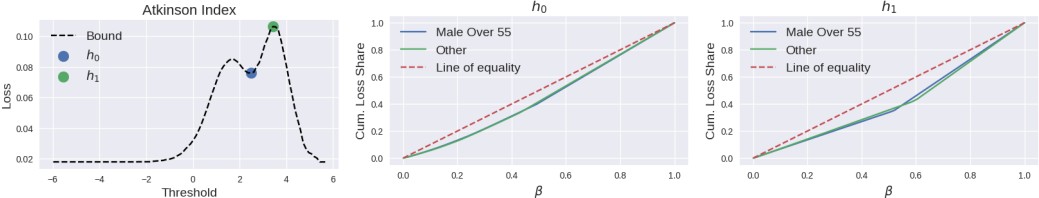

Figure 3: We select two hypotheses $h_0$ and $h_1$ with different bounds on Atkinson index produced using 2000 validation samples, and once again visualize the Lorenz curves induced by each. Tighter control on the Atkinson index leads to a more equal distribution of the loss (especially across the middle of the distribution, which aligns with the choice of $\epsilon$), highlighting the utility of being able to target such a metric in conservative model selection.

## 6 Related work

The field of distribution-free uncertainty quantification has its roots in conformal prediction [27]. The coverage guarantees of conformal prediction have recently been extended and generalized to controlling the expected loss of loss functions beyond coverage [1, 3]. The framework proposed by [30] offers the ability to select predictors beyond expected loss, to include a rich class of quantile-based risk measures (QBRMs) like CVaR and intervals of the VaR; they also introduce a method for achieving tighter bounds on certain QBRMs by focusing the statistical power of the Berk-Jones bound on a certain quantile range. Note that these measures cannot cover the range of dispersion measures studied in this work.

There is a rich literature studying both standard and group-based statistical dispersion measures, and their use in producing fairer outcomes in machine learning systems. Some work in fairness has aimed at achieving coverage guarantees across groups [24, 25], but to our knowledge there has not been prior work exploring controlling loss functions beyond coverage, such as the plethora of loss functions aimed at characterizing fairness, which can be expressed as group-based measures (cf. Section 3.2). Other recent fairness work has adapted some of the inequality measures found in economics. [32] aims to enforce that outcomes are not too different across groups defined by protected attributes, and introduces a convex notion of group CVaR, and [24] propose a DFUQ method of equalizing coverage between groups. [19] studies distributional inequality measures like Gini and Atkinson index since demographic group information is often unavailable, while [7] use the notion of Lorenz efficiency to generate rankings that increase the utility of both the worst-off users and producers.

## 7 Conclusion

In this work, we focus on a rich class of statistical dispersion measures, both standard group-based, and show how these measures can be controlled. In addition, we offer a novel numerical optimization method for achieving tighter bounds on these quantities. We investigate the effects of applying our framework via several experiments and show that our methods lead to more fair model selection and tighter bounds. We believe our study offers a significant step towards the sort of thorough and transparent validation that is critical for applying machine learning algorithms to applications with societal implications.

## Acknowledgements

We thank the Google Cyber Research Program and ONR (Award N00014-23-1-2436) for their generous support. J. Snell gratefully acknowledges financial support from the Schmidt DataX Fund at Princeton University made possible through a major gift from the Schmidt Futures Foundation.

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

# Appendix

**Broader impact.** The broader impact of the proposed framework is significant, as it extends the ability to gain trust in machine learning systems. However there are important concerns and limitations.

- **Focus on performance metrics** In this paper we propose a range of performance metrics, which extend well beyond standard metrics concerning expected loss. However, in many situations these metrics are not sufficient to capture the effects of the machine learning system. Often a number of different metrics are required to provide a clearer picture of model performance, while some effects are difficult to capture in any metric. Also, while the measures studied offer the ability to more evenly distribute a quantity across a population, they do not offer guarantees to individuals. Finally, achieving a more equal distribution of the relevant quantity (e.g., loss or income) may have negative impacts on some segments of the population.

- **Limitations** These are summarized in the Conclusion but are expanded upon here. An important assumption in this work, and in distribution-free uncertainty quantification more generally, is that the examples seen in deployment are drawn from the same distribution as those in the validation set that are used to construct the bounds. Although this is an active area of research, here we make this assumption, and the quality of the bounds produced may degrade if the assumption is violated. A second limitation is that the scope of hypotheses and predictors we can select from is limited, due to theoretical constraints: a correction must be performed based on the size of the hypothesis set. Finally, the generated bounds may not be tight, depending on the amount of available validation data and unavoidable limits of the techniques used to produce the bounds. We did some comparisons to Empirical values of the measures we obtained bounds for in the experiments; more extensive studies would be useful to elucidate the value of the bounds in practice.

**Organization of the Appendix.** (1) In Appendix A, we provide detailed statements and derivations of our methodology presented Section 4.2.1, including how to obtain bounds for those measures mentioned in Section 3; (2) in Appendix B, we introduce further societal dispersion measures, beyond those presented in Section 3 and corresponding bounds; (3) in Appendix C, we investigate the extension of our results to multi-dimensional settings; (4) lastly, in Appendix D and E, we provide more complete details and results from our experiments (Section 5).

## A   Derivations and proofs for bounding methods

Section A.1, we first consider how to control, or provide upper bounds on, various quantities when we are given $(\hat{F}_{n,L}^{\delta}, \hat{F}_{n,U}^{\delta})$, which are constructed by $\{X_i\}_{i=1}^n$, such that

$$\mathbb{P}(\hat{F}_{n,L}^{\delta,-} \preceq F \preceq \hat{F}_{n,U}^{\delta,-}) \geq 1 - \delta$$

where the randomness is taken over $\{X_i\}_{i=1}^n$.

Then, in Section A.2, we will show how we obtain $(\hat{F}_{n,L}^{\delta,-}, \hat{F}_{n,U}^{\delta,-})$ by extending the arguments in [30]. In addition, we show details in Section A.2.2 on how we go beyond the methods in [30] and provide a numerical optimization method for tighter bounds.

**Proof of Proposition 1.** We briefly describe the the proof for Proposition 1. The proof is mainly based on [30], but we include it here for completeness. Notice for any non-decreasing function $G : \mathbb{R} \to \mathbb{R}$ (not just a CDF), there exists the (general) inverse of $G$ as $G^-(p) = \inf\{x : G(x) \geq p\}$ for any $p \in \mathbb{R}$.

**Proposition 2** (Restatement of Proposition 1). *For the CDF $F$ of $X$, if there exists two increasing functions $F_U, F_L$ such that $F_U \succeq F \succeq F_L$, then we have $F_L^- \succeq F^- \succeq F_U^-$.*

*Proof.* For any two non-decreasing function $G(p)$ and $C(p)$, by the definition of the general inverse function, $G(G^-(p)) \geq p$. If $C \succeq G$, we therefore have $C(G^-(p)) \geq G(G^-(p)) \geq p$. Applying $C^-$ to both sides yields $C^-(C(G^-(p))) \geq C^-(p)$. But $x \geq C^- \circ C(x)$ (see e.g. Proposition 3 on p. 6

of [28]) and thus $G^-(p) \geq C^-(p)$. Plugging in $F$ and $F_U$ as $G$ and $C$, this can yield $F^- \succeq F_U^-$. The other direcion is similar. □

## A.1 Control of nonlinear functions of CDFs (Section 4.2.1)

### A.1.1 Control for monotonic functions

Recall that we start with the simplest case where $\xi$ is a monotonic function in the range of $X$. It is straightforward to have the following claim.

**Claim 1.** *If we have $\hat{F}_{n,L}^{\delta,-} \preceq F \preceq \hat{F}_{n,U}^{\delta,-}$ with probability at least $1 - \delta$ for some $\delta \in (0,1)$, if $\xi$ is an increasing function, then*

$$\xi(\hat{F}_{n,L}^{\delta,-}) \succeq \xi(\hat{F}^-) \succeq \xi(\hat{F}_{n,U}^{\delta,-})$$

*with probability at least $1 - \delta$. Similarly, if $\xi$ is a decreasing function, then $\xi(\hat{F}_{n,L}^{\delta,-}) \preceq \xi(\hat{F}^-) \preceq \xi(\hat{F}_{n,U}^{\delta,-})$ with probability at least $1 - \delta$.*

We show how this could be applied to provide bounds for Gini coefficient and Atkinson index by controlling the numerator and denominator separately as integrals of monotonic functions of $F^-$.

**Example 3** (Gini coefficient). *If given a $(1 - \delta)$-CBP $(\hat{F}_{n,L}^{\delta}, \hat{F}_{n,U}^{\delta})$ and $\hat{F}_{n,L}^{\delta} \succeq 0$ [4], we can provide the following bound for the Gini coefficient. Notice that*

$$\mathcal{G}(X) = \frac{\int_0^1 (2p - 1)F^-(p)dp}{\int_0^1 F^-(p)dp} = \frac{\int_0^1 2pF^-(p)dp}{\int_0^1 F^-(p)dp} - 1.$$

*Given $F^-(p) \geq 0$ (since we only consider non-negative losses, i.e. $X$ is always non-negative), we know*

$$\mathcal{G}(X) \leq \frac{\int_0^1 2p\hat{F}_{n,L}^{\delta,-}(p)dp}{\int_0^1 \hat{F}_{n,U}^{\delta,-}(p)dp} - 1,$$

*with probability at least $1 - \delta$.*

**Example 4** (Atkinson index). *First, we present the complete version of Atkinson index. Namely,*

$$\mathcal{A}(\varepsilon, X) := \begin{cases} 1 - \dfrac{\left( \int_0^1 (F^-(p))^{1-\varepsilon} dp \right)^{\frac{1}{1-\varepsilon}}}{\int_0^1 F^-(p)dp}, & \text{if } \varepsilon \geq 0, \ \varepsilon \neq 1; \\ 1 - \dfrac{\exp(\int_0^1 \ln(F^-(p))dp)}{\int_0^1 F^-(p)dp}, & \text{if } \varepsilon = 1. \end{cases}$$

*Notice that for $\varepsilon \geq 0$, $(\cdot)^{1-\varepsilon}$ and $\ln(\cdot)$ are increasing functions, thus, for Atkinson index and a $(1-\delta)$-CBP $(\hat{F}_{n,L}^{\delta}, \hat{F}_{n,U}^{\delta})$, if $\hat{F}_{n,L}^{\delta} \succeq 0$, let us define $\mathcal{A}_U^{\delta}(\varepsilon, X) := 1 - \dfrac{\left( \int_0^1 (\hat{F}_{n,U}^{\delta,-}(p))^{1-\varepsilon} dp \right)^{\frac{1}{1-\varepsilon}}}{\int_0^1 \hat{F}_{n,L}^{\delta,-}(p)dp}$, if $\varepsilon \geq 0, \varepsilon \neq 1$; $1 - \dfrac{\exp(\int_0^1 \ln(\hat{F}_{n,U}^{\delta,-}(p))dp)}{\int_0^1 \hat{F}_{n,L}^{\delta,-}(p)dp}$, if $\varepsilon = 1$. Then, with probability at least $1 - \delta$, $\mathcal{A}_U^{\delta}(\varepsilon, X)$ is an upper bound for $\mathcal{A}(\varepsilon, X)$ for all $\varepsilon \in [0, 1)$.*

As mentioned in Remark 1, instead of calculating bounds separately for each $\varepsilon$, simple post-processing enables us to efficiently issue a family of bounds.

**Example 5** (CVaR fairness-risk measures and beyond). *Recall that for $\alpha \in (0, 1)$,*

$$\mathcal{D}_{CV,\alpha}(T(F_g)) = \min_{\rho \in \mathbb{R}} \left\{ \rho + \frac{1}{1 - \alpha} \cdot \mathbb{E}_{g \sim \mathcal{P}_{Idx}}[T(F_g) - \rho]_+ \right\} - \mathbb{E}_{g \sim \mathcal{P}_{Idx}}[T(F_g)].$$

*The function $[T(F_g) - \rho]_+$ is an increasing function when $\rho$ is fixed and its further composition with the expectation operation is still increasing. If we have $(T_L^{\delta}(F_g), T_U^{\delta}(F_g))$ such that $T_L^{\delta}(F_g) \leq$*

---

[4]This can be easily achieved by taking truncation over 0. Also, the construction of $\hat{F}_{n,L}^{\delta}$ in Section A.2 always satisies this constraint.

$T(F_g) \le T_U^\delta(F_g)$ [5] *for all g with probability at least* $1 - \delta$, *then we have*

$$\mathcal{D}_{CV,\alpha}(T(F_g)) \le \min_{\rho \in \mathbb{R}} \left\{ \rho + \frac{1}{1-\alpha} \cdot \mathbb{E}_{g \sim \mathcal{P}_{Idx}}[T_U^\delta(F_g) - \rho]_+ \right\} - \mathbb{E}_{g \sim \mathcal{P}_{Idx}}[T_L^\delta(F_g)],$$

*and the first term of RHS can be minimized easily since it is a convex function of* $\rho$.

### A.1.2 Control for absolute and polynomial functions

Recall that if $s_L \le s \le s_U$, then

$$s_L \mathbf{1}\{s_L \ge 0\} - s_U \mathbf{1}\{s_U \le 0\} \le |s| \le \max\{|s_U|, |s_L|\}.$$

More generally, for any polynomial function $\phi(s) = \sum_{k=0} \alpha_k s^k$. Notice if $k$ is odd, $s^k$ is monotonic w.r.t. $s$ and we can bound

$$\phi(s) \le \sum_{\{k \text{ is odd}, \ \alpha_k \ge 0\}} \alpha_k s_U^k + \sum_{\{k \text{ is odd}, \ \alpha_k < 0\}} \alpha_k s_L^k$$
$$+ \sum_{\{k \text{ is even}, \ \alpha_k \ge 0\}} \alpha_k \max\{|s_L|^k, |s_U|^k\} + \sum_{\{k \text{ is even}, \ \alpha_k < 0\}} \alpha_k (s_L \mathbf{1}\{s_L \ge 0\} - s_U \mathbf{1}\{s_U \le 0\})^k.$$

So, for $\phi(F^-)$, we can plug in $\hat{F}_{n,L}^{\delta,-}$ and $\hat{F}_{n,U}^{\delta,-}$ to replace $s_U$ and $s_L$ to obtain an upper bound with probability at least $(1 - \delta)$. The derivation for the lower bound is similar. We summarize our results as the following proposition.

**Proposition 3.** *If given a* $(1 - \delta)$-*CBP, then with probability at least* $1 - \delta$, $(\hat{F}_{n,L}^\delta, \hat{F}_{n,U}^\delta)$,

$$\hat{F}_{n,U}^{\delta,-} \mathbf{1}\{\hat{F}_{n,U}^{\delta,-} \ge 0\} - \hat{F}_{n,L}^{\delta,-} \mathbf{1}\{\hat{F}_{n,L}^{\delta,-} \le 0\} \preceq |F^-| \preceq \max\{|\hat{F}_{n,L}^{\delta,-}|, |\hat{F}_{n,}^{\delta,-}|\}.$$

*Moreover, for any polynomial function* $\phi(s) = \sum_{k=0} \alpha_k s^k$, *we have*

$$\phi(F^-) \preceq \sum_{\{k \text{ is odd}, \ \alpha_k \ge 0\}} \alpha_k (\hat{F}_{n,L}^{\delta,-})^k + \sum_{\{k \text{ is odd}, \ \alpha_k < 0\}} \alpha_k (\hat{F}_{n,U}^{\delta,-})^k$$
$$+ \sum_{\{k \text{ is even}, \ \alpha_k \ge 0\}} \alpha_k \max\{|\hat{F}_{n,U}^{\delta,-}|^k, |\hat{F}_{n,L}^{\delta,-}|^k\}$$
$$+ \sum_{\{k \text{ is even}, \ \alpha_k < 0\}} \alpha_k (\hat{F}_{n,U}^{\delta,-} \mathbf{1}\{\hat{F}_{n,U}^{\delta,-} \ge 0\} - \hat{F}_{n,L}^{\delta,-} \mathbf{1}\{\hat{F}_{n,L}^{\delta,-} \le 0\})^k.$$

**Example 6.** *If we have* $(T_L^\delta(F_g), T_U^\delta(F_g))$ *such that* $T_L^\delta(F_g) \le T(F_g) \le T_U^\delta(F_g)$ *holds for all g we consider, then we can provide high probability upper bounds for*

$$\xi(T(F_{g_1}) - T(F_{g_2}))$$

*for any polynomial functions or the absolute function* $\xi$. *For example, with probability at least* $1 - \delta$,

$$|T(F_{g_1}) - T(F_{g_2})| \le \max\{|T_U^\delta(F_{g_1}) - T_L^\delta(F_{g_2})|, |T_L^\delta(F_{g_1}) - T_U^\delta(F_{g_2})|\}.$$

We will further show in Appendix B how our results are applied to specific examples.

### A.1.3 Control for a general function

To handle general non-linearity, we need to introduce the class of functions of bounded variation on a certain interval, which is a very rich class that includes all the functions that are continuously differentiable or Lipchitz continuous on that interval.

**Definition 4** (Functions of bounded total variation [26])**.** *Define the set of paritions on* $[a, b]$ *as*

$$\Pi = \{\pi = (x_0, x_1, \cdots, x_{n_\pi}) \mid \pi \text{ is a partition of } [a, b] \text{ satisfying } x_i \le x_{i+1} \text{ for all } 0 \le i \le n_\pi - 1\}.$$

---

[5]$T(F_g)$ here is one of the functionals in the form we studied, so that we can provide upper and lower bounds for it.

Then, the total variation of a continuous real-valued function $\xi$, defined on $[a, b] \subset \mathbb{R}$ is defined as

$$V_a^b(\xi) := \sup_{\pi \in \Pi} \sum_{i=0}^{n_\pi} |\xi(x_{i+1}) - \xi(x_i)|$$

where $\Pi$ is the set of all partitions, and we say a function $\xi$ is of bounded variation, i.e. $\xi \in BV([a, b])$ iff $V_a^b(\xi) < \infty$.

Recall that $X \geq 0$ in our cases, then, for $\xi(F^-)$, we can have the following bound.

**Theorem 2** (A restatement & formal version of Theorem 1). *For a* $(1 - \delta)$-*CBP* $(\hat{F}_{n,L}^\delta, \hat{F}_{n,U}^\delta)$, *for any* $p \in [0, 1]$ *such that the total variation of* $\xi$ *is finite on* $[0, \hat{F}_{n,L}^{\delta,-}(p)]$, *then*

$$\xi(F^-(p)) \leq V_0^{\hat{F}_{n,L}^{\delta,-}(p)}(\xi) - V_0^{\hat{F}_{n,U}^{\delta,-}(p)}(\xi) + \xi(\hat{F}_{n,U}^{\delta,-}(p)).$$

*Moreover, if* $\xi$ *is continuously differentiable on* $[0, \hat{F}_{n,L}^{\delta,-}(p)]$, *we can express* $V_0^s(\xi)$ *as* $\int_0^x |\frac{d\xi}{ds}(s)| ds$ *for any* $x \in [0, \hat{F}_{n,L}^{\delta,-}(p)]$.

*Proof.* By the property of functions of bounded total variation [26], if $\xi$ is of bounded total variation on $[0, \hat{F}_{n,L}^{\delta,-}(p)]$, then, we have that: for any $x \in [0, \hat{F}_{n,L}^{\delta,-}(p)]$

$$\xi(x) = V_0^x(\xi) - (V_0^x(\xi) - \xi(x))$$

where both $f_1(x) := V_0^x(\xi)$ and $f_2(x) := V_0^x(\xi) - \xi(x)$ are increasing functions. Moreover,

$$V_0^x(\xi) = \int_0^x \left| \frac{d\xi}{ds}(s) \right| ds$$

if $\xi$ is continuously differentiable.

Thus, by taking advantage of the monotonicity, we have

$$\xi(F^-(p)) \leq V_0^{\hat{F}_{n,L}^{\delta,-}(p)}(\xi) - V_0^{\hat{F}_{n,U}^{\delta,-}(p)}(\xi) + \xi(\hat{F}_{n,U}^{\delta,-}(p)).$$

So, if $\xi$ is of bounded variation on the range of $X$, then

$$\xi(F^-) \preceq V_0^{\hat{F}_{n,L}^{\delta,-}}(\xi) - V_0^{\hat{F}_{n,U}^{\delta,-}}(\xi) + \xi(\hat{F}_{n,U}^{\delta,-}) = f_1(\hat{F}_{n,L}^{\delta,-}) - f_2(\hat{F}_{n,U}^{\delta,-}).$$

$\square$

## A.2   Methods to obtain confidence two-sided bounds for CDFs (Section 4.1)

We provide details for two-sided bounds and our numerical methods in the following.

### A.2.1   The reduction approach to constructing upper bounds of CDFs (Section 4.1.1)

We here provide the proof of Lemma 1.

**Lemma 2** (A restatement & formal version of Lemma 1). *For* $0 \leq L_1 \leq L_2 \cdots \leq L_n \leq 1$, *since* $\mathbb{P}(\forall i : F(X_{(i)}) \geq L_i) \geq \mathbb{P}(\forall i : U_{(i)} \geq L_i)$ *by [30], if we further have* $\mathbb{P}(\forall i : U_{(i)} \geq L_i) \geq 1 - \delta$, *then we have*

$$\mathbb{P}(\forall i : \lim_{\epsilon \to 0^+} F(X_{(i)} - \epsilon) \leq 1 - L_{n-i+1}) \geq 1 - \delta.$$

*Furthermore, let* $R(x)$ *be defined as*

$$R(x) = \begin{cases} 1 - L_n, & \text{for } x < X_{(1)} \\ 1 - L_{n-1}, & \text{for } X_{(1)} \leq x < X_{(2)} \\ \cdots \\ 1 - L_1, & \text{for } X_{(n-1)} \leq x < X_{(n)} \\ 1, & \text{for } X_{(n)} \leq x. \end{cases}$$

*Then,* $F \preceq R$.

*Proof.* Notice that for given order statistics $\{X_{(i)}\}_{i=1}^n$, let $\mathbb{P}_{\{X_{(i)}\}_{i=1}^n}$ denote the probability taken over the randomness of $\{X_{(i)}\}_{i=1}^n$, and $\mathbb{P}_X$ denote the probability taken over the randomness of $X$, which is an independent random variable drawn from $F$. Let us denote $B = -X$, and $B_{(i)}$ as the $i$-th order statistic for samples $\{-X_i\}_{i=1}^n$. It is easy to see that $B_{(n-i+1)} = -X_{(i)}$. We also denote $\mathbb{P}_B$ as the probability taken over the randomness of $B$, and $F_B$ as the CDF of $B$.

$$
\begin{aligned}
\mathbb{P}_{\{X_{(i)}\}_{i=1}^n}(\forall i: \lim_{\epsilon \to 0^+} F(X_{(i)} - \epsilon) \le 1 - L_{n-i+1}) &= \mathbb{P}_{\{X_{(i)}\}_{i=1}^n}(\forall i: \mathbb{P}_X(X \ge X_{(i)}) > L_{n-i+1}) \\
&= \mathbb{P}_{\{X_{(i)}\}_{i=1}^n}(\forall i: \mathbb{P}_X(-X \le -X_{(i)}) > L_{n-i+1}) \\
&= \mathbb{P}_{\{X_{(i)}\}_{i=1}^n}(\forall i: \mathbb{P}_B(B \le B_{(n-i+1)}) > L_{n-i+1}) \\
&= \mathbb{P}(\forall i: F_B \circ F_B^-(U_{(n-i+1)}) > L_{n-i+1}) \\
&\ge \mathbb{P}(\forall i: U_{(n-i+1)} > L_{n-i+1}).
\end{aligned}
$$

where we use the fact that $F_B^-(U_{(n-i+1)})$ is of the same distribution as $B_{(n-i+1)}$ and the last inequality follows from Proposition 1, eq. 24 on p.5 of [28].

Notice that $\mathbb{P}(\forall i: U_{(n-i+1)} > L_{n-i+1}) = \mathbb{P}(\forall i: U_{(n-i+1)} \ge L_{n-i+1})$, and according to [30] and our assumption, $\mathbb{P}(\forall i: F(X_{(i)}) \ge L_i) \ge \mathbb{P}(\forall i: U_{(i)} \ge L_i) \ge 1 - \delta$.

The conservative construction of $R$ satisfies $R \succeq F$ straightforwardly if $\forall i: \lim_{\epsilon \to 0^+} F(X_{(i)} - \epsilon) \le 1 - L_{n-i+1})$ holds. Thus, we know $R \succeq F$ with probability at least $1 - \delta$. Our proof is complete. $\square$

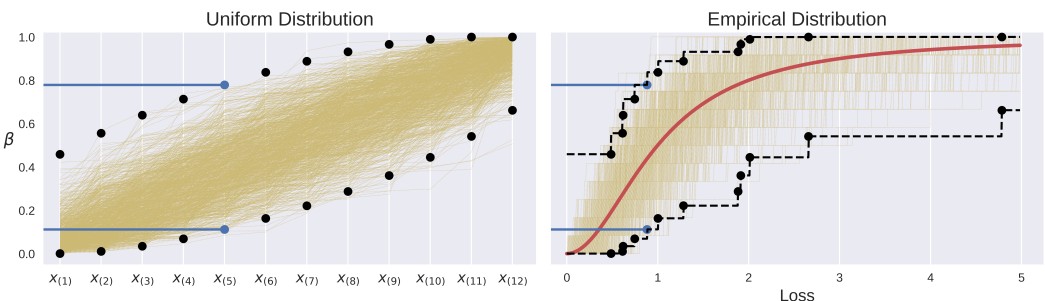

Figure 4: Example illustrating the construction of distribution-free CDF lower and upper bounds by bounding order statistics. On the left, order statistics are drawn from a uniform distribution. On the right, samples are drawn from a real loss distribution, and the corresponding Berk-Jones CDF lower and upper bound are shown in black. Our distribution-free method gives bound $b_i^{(l)}$ and $b_i^{(u)}$ on each sorted order statistic such that the bound depends only on $i$, as illustrated in the plots for $i = 5$ (shown in blue). On the left, 1000 realizations of $x_{(1)}, \ldots, x_{(n)}$ are shown in yellow. On the right, 1000 empirical CDFs are shown in yellow, and the true CDF $F$ is shown in red.

### A.2.2 Details of numerical optimization method (Section 4.3)

Now, we introduce the details of our numerical optimization method. Recall that one drawback of the QBRM bounding approach is that it is not weight function aware: when controlling $\int_0^1 \psi(p) F^-(p) dp$ for a non-negative weight function $\psi$, the procedure ignores the structure of $\psi$, as it first obtains $\hat{F}_{n,L}^\delta$, then provides an upper bound $\int_0^1 \psi(p) \hat{F}_{n,L}^{\delta,-}(p) dp$.

Our numerical approach can overcome that drawback and can also easily be applied to handle mixtures of multiple functionals. The bounds obtained by our method are significantly tighter than those provided by methods in [30] in the regime of small data size. Notice that the small data size regime is the one people care about because when the data size is large, all the bounds we discussed will converge to the same value, and the gap between different bounds will shrink to 0 as the data size grows.

First, by [**?** ] and Proposition 1, eq. 24 on p.5 of [28], we have for any $0 \leq L_1 \leq \cdots \leq L_n \leq 1$,

$$\mathbb{P}\big(\forall i, \ F(X_{(i)}) \geq L_i\big) \geq \mathbb{P}\big(\forall i, \ U_{(i)} \geq L_i\big)$$

$$\geq n! \int_{L_n}^{1} dx_n \int_{L_{n-1}}^{x_n} dx_{n-1} \cdots \int_{L_1}^{x_2} dx_1,$$

where the right-hand side integral is a function of $\{L_i\}_{i=1}^{n}$ and its partial derivatives can be exactly calculated by the package in [21]. Specifically, the package in [21] enables us to calculate

$$v(L_1, L_2, \cdots, L_n, 1) := \int_{L_n}^{1} dx_n \int_{L_{n-1}}^{x_n} dx_{n-1} \cdots \int_{L_1}^{x_2} dx_1$$

for any positive integer $n$. Notice that the partial derivative of $v(L_1, L_2, \cdots, L_n, 1)$ with respect to $L_i$ is:

$$\partial_{L_i} v(L_1, L_2, \cdots, L_n, 1) = - \int_{L_n}^{1} dx_n \int_{L_{n-1}}^{x_n} dx_{n-1} \cdots \int_{L_{i+1}}^{x_{i+2}} dx_{i+1}$$

$$\cdot \int_{L_{i-1}}^{L_i} dx_{i-1} \cdots \int_{L_1}^{x_2} dx_1,$$

$$= -v(L_{i+1}, \cdots, L_n, 1) \cdot v(L_1, \cdots, L_{i-1}, L_i),$$

which we can also use the package in [21] to calculate the partial derivatives.

Consider providing upper or lower bounds for $\int_0^1 \psi(p) F^-(p) dp$ for non-negative weight function $\psi$ as an example. For any $\{L_i\}_{i=1}^{n}$ satisfying $\mathbb{P}\big(\forall i, \ F(X_{(i)}) \geq L_i\big) \geq 1 - \delta$, one can use conservative CDF completion in [30] to obtain $\hat{F}_{n,L}^{\delta}$, i.e. $\int_0^1 \psi(p)\xi(\hat{F}_{n,L}^{\delta,-}(p))dp = \sum_{i=1}^{n+1} \xi(X_{(i)}) \int_{L_{i-1}}^{L_i} \psi(p) dp$, where $L_{n+1}$ is 1, $L_0 = 0$, and $X_{(n+1)} = \infty$ or a known upper bound for $X$. Then, we can formulate tightening the upper bound as an optimization problem:

$$\min_{\{L_i\}_{i=1}^{n}} \sum_{i=1}^{n+1} \xi(X_{(i)}) \int_{L_{i-1}}^{L_i} \psi(p) dp$$

such that

$$\mathbb{P}\big(\forall i, \ F(X_{(i)}) \geq L_i\big) \geq 1 - \delta, \text{ and } 0 \leq L_1 \leq \cdots \leq L_n \leq 1.$$

Similarly, for the lower bound, we can use the CDF completion mentioned in Theorem 1, and construct $\hat{F}_{n,U}^{\delta}$, then, we can study the following lower bound for $\int_0^1 \psi(p) F^-(p) dp$,

$$\sum_{i=1}^{n} \xi(X_{(i)}) \int_{L_{n-i}}^{L_{n-i+1}} \psi(p) dp$$

where $X_{(0)} = 0$.

**Parameterized model approach.** Notice the above optimization problem formulation has a drawback: if more samples are drawn, i.e. $n$ increases, then the number of parameters we need to optimize also increases. In practice, we re-parameterize $\{L_i\}_{i=1}^{n}$ as the following:

$$L_i(\theta) = \frac{\sum_{j=1}^{i} \exp(\phi_\theta(g_j))}{1 + \sum_{j=1}^{n} \exp(\phi_\theta(g_j))}$$

where $g_i$ are random Gaussian seeds. This is of the same spirit as using random seeds in generative models. We find that a simple parameterized neural network model with 3 fully-connected hidden layers of dimension 64 is enough for good performance and robust to hyper-parameter settings. Take the upper bound optimization problem as an example; using the new parameterized model, we have

$$\min_{\{\theta\}_{i=1}^{n}} \sum_{i=1}^{n+1} \xi(X_{(i)}) \int_{L_{i-1}(\theta)}^{L_i(\theta)} \psi(p) dp \tag{1}$$

such that

$$n! \int_{L_n(\theta)}^1 dx_n \int_{L_{n-1}(\theta)}^{x_n} dx_{n-1} \cdots \int_{L_1(\theta)}^{x_2} dx_1 \geq 1 - \delta,$$

where $L_0 = 0$, $L_{n+1} = 1$, $X_{(n+1)} = \infty$ or a known upper bound for $X$. We can solve the above optimization problem using heuristic methods such as [9].

**Post-processing for a rigorous guarantee for constraints.** Notice that we may not ensure the constraint $n! \int_{L_n(\theta)}^1 dx_n \int_{L_{n-1}(\theta)}^{x_n} dx_{n-1} \cdots \int_{L_1(\theta)}^{x_2} dx_1 \geq 1 - \delta$ is satisfied in the above optimization because we may use surrogates like Langrange forms in our optimization processes. To make sure the constraint is strictly satisfied, we can do the following post-processing: let us denote the obtained $L_i$'s by optimizing (1) as $L_i(\hat\theta)$. Then, we look for $\gamma^* \in [0, L_n(\hat\theta)]$ such that

$$\gamma^* = \inf\{\gamma : n! \upsilon(L_1(\hat\theta) - \gamma, \cdots, L_n(\hat\theta) - \gamma, 1) \geq 1 - \delta, \gamma \geq 0\}.$$

Notice there is always a feasible solution as when $\gamma = L_n(\hat\theta)$,

$$n! \upsilon(L_1(\hat\theta) - \gamma, \cdots, L_n(\hat\theta) - \gamma, 1) \geq \mathbb{P}\big(\forall i, \ U_{(i)} \geq 0\big) = 1$$

and $\upsilon(L_1(\hat\theta) - \gamma, \cdots, L_n(\hat\theta) - \gamma, 1)$ is a decreasing function of $\gamma$. We can use binary search to efficiently find (a good approximate of) $\gamma^*$.

# B    Other dispersion measures and calculation

## B.1    Lorenz curve & the extended Gini family

**Lorenz curve.**    In the main context, Lorenz curve has been mentioned in reference to Gini coefficient and Atkinson index. To be more complete, we further demonstrate the definition of Lorenz curve in its mathematical form.

**Definition 5** (Lorenz curve). *The definition of Lorenz curve is a function: for $t \in [0, 1]$,*

$$\mathcal{L}(t) = \frac{\int_0^t F^{-1}(p)\, dp}{\int_0^1 F^{-1}(p)\, dp}.$$

We can obtain a lower bound and an upper bound function for the Lorenz curve. Given a $(1 - \delta)$-CBP $(\hat{F}_{n,L}^\delta, \hat{F}_{n,U}^\delta)$ and $\hat{F}_{n,L}^\delta \succeq 0$, we can construct a lower bound function $\mathcal{L}_L^\delta(t)$:

$$\mathcal{L}_L^\delta(t) = \frac{\int_0^t \hat{F}_{n,U}^{\delta,-}(p)\, dp}{\int_0^1 \hat{F}_{n,L}^{\delta,-}(p)\, dp},$$

and an upper bound can be obtained by

$$\mathcal{L}_U^\delta(x) = \frac{\int_0^t \hat{F}_{n,L}^{\delta,-}(p)\, dp}{\int_0^1 \hat{F}_{n,U}^{\delta,-}(p)\, dp}.$$

With probability at least $1 - \delta$, the true Lorenz curve sits between the upper bound function and the lower bound function for all $t \in [0, 1]$.

**The extended Gini family.**    The Gini coefficient can further give rise to the extended Gini family, which is a family of variability and inequality measures that depends on one parameter – the extended Gini parameter. The definition is as follows.

**Definition 6** (The extended Gini family[34]). *The extended Gini coefficient is given by*

$$\mathcal{G}(\nu, X) := \frac{-\nu Cov(X, [1 - F(X)]^{\nu - 1})}{\mathbb{E}[X]}$$

$$= 1 - \frac{\nu \int_0^1 (1 - p)^{\nu - 1} F^-(p) dp}{\int_0^1 F^-(p) dp},$$

*where $\nu > 0$ is the extended Gini parameter and $Cov(\cdot, \cdot)$ is the covariance.*

For the extended Gini coefficient, choosing different $\nu$'s corresponds to different weighting schemes applied to the vertical distance between the egalitarian line and the Lorenz curve; and if $\nu = 2$, it is the standard Gini coefficient.

Given a $(1-\delta)$-CBP $(\hat{F}_{n,L}^{\delta}, \hat{F}_{n,U}^{\delta})$ and $\hat{F}_{n,L}^{\delta} \succeq 0$, we can construct upper bound for $\mathcal{G}$. Let

$$\mathcal{G}_U^{\delta}(\nu, X) := 1 - \frac{\nu \int_0^1 (1-p)^{\nu-1} \hat{F}_{n,U}^{\delta,-}(p)dp}{\int_0^1 \hat{F}_{n,L}^{\delta,-}(p)dp},$$

then $\mathcal{G}_U^{\delta}(\nu, X) \succeq \mathcal{G}(\nu, X)$ with probability at least $1 - \delta$.

## B.2 Generalized entropy index

The generalized entropy index [29] is another measure of inequality in a population. Specifically, the definition is: for real number $\alpha$

$$GE(\alpha, X) := \begin{cases} \frac{1}{\alpha(\alpha-1)} \mathbb{E}\left[ \left( \frac{X}{\mathbb{E}X} \right)^{\alpha} - 1 \right], & \alpha \neq 0, 1 \\ \mathbb{E}\left[ \frac{X}{\mathbb{E}X} \ln(\frac{X}{\mathbb{E}X}) \right], & \text{if } \alpha = 1 \\ -\mathbb{E}\left[ \ln(\frac{X}{\mathbb{E}X}) \right], & \text{if } \alpha = 0. \end{cases}$$

It is not hard to further expand the expressions and write the generalized entropy index as:

$$GE(\alpha, X) := \begin{cases} \frac{1}{\alpha(\alpha-1)} \int_0^1 \left[ \left( \frac{F^-(p)}{\int_0^1 F^-(p)dp} \right)^{\alpha} - 1 \right] dp, & \alpha \neq 0, 1 \\ \int_0^1 \left[ \frac{F^-(p)}{\int_0^1 F^-(p)dp} \ln(\frac{F^-(p)}{\int_0^1 F^-(p)dp}) \right] dp, & \text{if } \alpha = 1 \\ -\int_0^1 \left[ \ln(\frac{F^-(p)}{\int_0^1 F^-(p)dp}) \right] dp, & \text{if } \alpha = 0. \end{cases}$$

Notice that $(\cdot)^{\alpha}$ is a monotonic function for the case $\alpha \neq 0, 1$, and $\ln(\cdot)$ is also a monotonic function, so the bound can be obtained similarly as in the case of Atkinson index. For instance, for $\alpha > 1$, given a $(1-\delta)$-CBP $(\hat{F}_{n,L}^{\delta}, \hat{F}_{n,U}^{\delta})$,

$$\frac{1}{\alpha(\alpha-1)} \int_0^1 \left[ \left( \frac{F^-(p)}{\int_0^1 F^-(p)dp} \right)^{\alpha} - 1 \right] dp \leq \frac{1}{\alpha(\alpha-1)} \int_0^1 \left[ \left( \frac{\hat{F}_{n,L}^{\delta,-}(p)}{\int_0^1 \hat{F}_{n,U}^{\delta,-}(p)dp} \right)^{\alpha} - 1 \right] dp.$$

Other cases can be tackled in a similar way, which we will not reiterate here.

## B.3 Hoover index

The Hoover index [16] is equal to the percentage of the total population's income that would have to be redistributed to make all the incomes equal.

**Definition 7** (Hoover index). *For a non-negative random variable $X$, the Hoover index is defined as*

$$H(X) = \frac{\int_0^1 |F^-(p) - \int_0^1 F^-(q)dq| dp}{2 \int_0^1 F^-(p)dp}$$

Hoover index involves forms like $|F^- - \mu|$ for $\mu = \int_0^1 F^-(p)dp$. This type of nonlinear structure can be dealt with the absolute function results mentioned in Appendix A.1.2.

For Hoover index and a $(1-\delta)$-CBP $(\hat{F}_{n,L}^{\delta}, \hat{F}_{n,U}^{\delta})$, let us define

$$H_U(X) = \frac{\int_0^1 \max\{|\hat{F}_{n,L}^{\delta,-}(p) - \int_0^1 \hat{F}_{n,U}^{\delta,-}(q)dq|, |\hat{F}_{n,U}^{\delta,-}(p) - \int_0^1 \hat{F}_{n,L}^{\delta,-}(q)dq|\} dp}{2 \int_0^1 \hat{F}_{n,U}^{\delta,-}(p)dp}.$$

Then, with probability at least $1 - \delta$, $H_U(, X)$ is an upper bound for $H(X)$.

## B.4 Extreme observations & mean range

For example, a city may need to estimate the cost of damage to public amenities due to rain in a certain month. The loss for each day of a month is $X_1, \cdots, X_k$ i.i.d drawn from $F$, and the administration hopes to estimate and control the dispersion of the losses in a month so that they can accurately allocate resources. This involves quantities such as range ($\max_{i \in [k]} X_i - \min_{j \in [k]} X_j$) or quantiles of extreme observations ($\max_{i \in [k]} X_i$). The CDF of extreme observations such as $\max_{i \in [k]} X_i$ involves a nonlinear function of $F$, i.e. $(F(x))^k$.

**Example 7** (Quantiles of extreme observations). *The CDF of $\max_{i \in [k]} X_i$ is $F^k$. Thus, by the result of Appendix A.1.2, if given a $(1 - \delta)$-CBP $(\hat{F}^\delta_{n,L}, \hat{F}^\delta_{n,U})$ and $1 \succeq \hat{F}^{\delta,-}_{n,U} \succeq \hat{F}^{\delta,-}_{n,L} \succeq 0$, with probability at least $1 - \delta$,*

$$(\hat{F}^{\delta,-}_{n,L})^k \preceq F^k \preceq (\hat{F}^{\delta,-}_{n,U})^k.$$

*We also have*

$$(\hat{F}^{\delta,-}_{n,U})^k \preceq F^k \preceq (\hat{F}^{\delta,-}_{n,L})^k.$$

*Similarly, for $\min_{i \in [k]} X_i$, the CDF is $1 - (1 - F)^k$, thus, we have*

$$1 - (1 - \hat{F}^{\delta,-}_{n,U})^k \preceq F^k \preceq 1 - (1 - \hat{F}^{\delta,-}_{n,L})^k.$$

*We also want to emphasize, even if, $X$ is **not** necessarily non-negative, we can apply the polynomial method in Appendix A.1.2 for $\hat{F}^{\delta,-}_{n,U}$ and $\hat{F}^{\delta,-}_{n,L}$.*

**Example 8** (Mean range). *By [13], if we further have prior knowledge that $X$ is of continuous distribution, the mean of $\max_{i \in [k]} X_i - \min_{j \in [k]} X_j$ can be expressed as:*

$$k \int F^-(x)[F^{k-1}(x) - F^k(x)]dF(x) = k \int_0^1 F^-(F^-(p))[F^{k-1}(F^-(p)) - F^k(F^-(p))]dp$$

*Notice that both $F$ and $F^-$ are increasing. Thus, if given a $(1 - \delta)$-CBP $(\hat{F}^\delta_{n,L}, \hat{F}^\delta_{n,U})$, $\hat{F}^\delta_{n,L} \succeq 0$, then with probability at least $1 - \delta$,*

$$\int_0^1 \hat{F}^{\delta,-}_{n,L}(\hat{F}^{\delta,-}_{n,L}(p))\left[(\hat{F}^\delta_{n,U})^k(\hat{F}^{\delta,-}_{n,L}(p)) - (\hat{F}^\delta_{n,L})^k(\hat{F}^{\delta,-}_{n,U}(p))\right]dp$$

*is an upper bound of the mean range.*

There are many other interesting societal dispersion measures that could be handled by our framework, such as those in [19]. For example, they study tail share that captures "the top 1% of people own $X$ share of wealth", which could be easily handled with the tools provided here. We will leave those those examples to readers.

## C   Extension to multi-dimensional cases and applications

We briefly discuss extending our approach to multi-dimensional losses. Unfortunately, there is not a gold-standard definition of quantiles in the multi-dimensional case, and thus we only discuss functionals of CDFs and provide an example. For multi-dimensional samples $\{\boldsymbol{X}_i\}^n_{i=1}$, each of $k$ dimensions, i.e. $\boldsymbol{X}_i = (X_1^i, \cdots, X_k^i)$, for any $k$-dimensional vector $\boldsymbol{x} = (x_1, \cdots, x_k)$, define empirical CDF

$$\hat{F}_n(\boldsymbol{x}) = \frac{1}{n}\sum_{i=1}^n \mathbf{1}\{\boldsymbol{X}_i \preceq \boldsymbol{x}\}.$$

where we abuse the notation $\preceq$ to mean all of $\boldsymbol{X}_i$'s coordinates are smaller than $\boldsymbol{x}$'s.

By classic DKW inequality, we have with probability at least $1 - \delta$,

$$|\hat{F}_n(\boldsymbol{x}) - F(\boldsymbol{x})| \leq \sqrt{\frac{\ln(k(n+1)/\delta)}{2n}}.$$

Meanwhile, we can further adopt Frechet-Hoeffeding bound, which gives,

$$\max\{1 - k + \sum_{i=1}^k F_i(x_i), 0\} \leq F(\boldsymbol{x}) \leq \min\{F_1(x_1), \cdots, F_k(x_k)\}$$

where $F_i$ is the CDF of the i-th coordinate. Then, we can construct $(\hat{F}_{n,L}^{\delta/k,i}, \hat{F}_{n,U}^{\delta/k,i})$ such that $(\hat{F}_{n,L}^{\delta/k,i} \preceq F_i \preceq \hat{F}_{n,U}^{\delta/k,i})$, with probability at last $1 - \delta/k$. Thus, by union bound,

$$\max\{1 - k + \sum_{i=1}^{k} \hat{F}_{n,L}^{\delta/k,i}(x_i), 0\} \leq F(\boldsymbol{x}) \leq \min\{\hat{F}_{n,U}^{\delta/k,1}(x_1), \cdots, \hat{F}_{n,U}^{\delta/k,k}(x_k)\}$$

for all $\boldsymbol{x}$ with probability at last $1 - \delta$.

We have

$$F(\boldsymbol{x}) \geq \max\{1 - k + \sum_{i=1}^{k} \hat{F}_{n,L}^{\delta/k,i}(x_i), 0, \hat{F}_n(\boldsymbol{x}) - \sqrt{\frac{\ln(k(n+1)/\delta)}{2n}}\}$$

$$F(\boldsymbol{x}) \leq \min\{\hat{F}_{n,U}^{\delta/k,1}(x_1), \cdots, \hat{F}_{n,U}^{\delta/k,k}(x_k), \hat{F}_n(\boldsymbol{x}) + \sqrt{\frac{\ln(k(n+1)/\delta)}{2n}}\}$$

with probability at last $1 - 2\delta$.

**Example 9** (Gini correlation coefficient [34]). *The Gini correlation coefficient for two non-negative random variable $X$ and $Y$ are defined as*

$$\Gamma_{X,Y} := \frac{Cov(X, F_Y(Y))}{Cov(X, F_X(X))} = \frac{\int\int \left(F_{X,Y}(x,y) - F_X(x)F_Y(Y)\right) dx dF_Y(y)}{Cov(X, F_X(X))},$$

*where $F_X, F_Y$ are marginal CDFs of $X, Y$ and $F_{X,Y}$ is the joint CDF. One can use the multi-dimensional CDF bounds and our previous methods to provide bounds for the Gini correlation coeffiecient.*

# D   Experiment details

This section contains additional details for the experiments in Section 5. We set $\delta = 0.05$ (before statistical corrections for multiple tests) in all experiments unless otherwise explicitly stated. Whenever we are bounding measures on multiple hypotheses, we perform a correction for the size of the hypothesis set. Additionally, when we bound measures on multiple distributions (e.g. demographic groups), we also perform a correction. Our code will be released publicly upon the publication of this article.

## D.1   CivilComments (Section 5.1)

Our set of hypotheses are a toxicity model combined with a Platt scaler [22], where the model is fixed and we vary the scaling parameter in the range $[0.25, 2]$ while fixing the bias term to 0. We use a pre-trained toxicity model from the popular python library Detoxify [6] [10] and perform Platt Scaling using code from the python library released by [18] [7]. A Platt calibrator produces output according to:

$$h(v) = \frac{1}{1 + \exp(wv + b)}$$

where $w, b$ are learnable parameters and $v$ is the log odds of the prediction. Thus we form our hypothesis set by varying the parameter $w$ while fixing $b$ to 0. Examples are drawn from the train split of CivilComments, which totals 269,038 data points.

The loss metric for our CivilComments experiments is the Brier Score. For $n$ data points, Brier score is calculated as:

$$L = \frac{1}{n} \sum_{i=1}^{n} (f_i - o_i)^2$$

where $f_i$ is prediction confidence and $o_i$ is the outcome (0 or 1).

---

[6]`https://github.com/unitaryai/detoxify`
[7]`https://github.com/p-lambda/verified_calibration`

### D.1.1 Bounding complex objectives (Section 5.1.1)

We randomly sample 100,000 test points for calculating the empirical values in Table 1, and draw our validation points from the remaining data. We perform a Bonferroni correction on $\delta = 0.05$ for the size of the set of hypotheses as well as the number of distributions on which we bound our measures (in this case the number of groups, 4). We set $\lambda = 1.0$.

Numerical optimization details (including training strategy and hyperparameters) are the same as Section 5.1.2, explained below in Appendix D.1.2. For each group $g$ we optimize the objective

$$\mathcal{O} = T_1(F_g) + T_2(F_g)$$

where $F_g$ is the CDF bound for group, $T_1$ is expected loss, and $T_2$ is a smoothed version of a median with $a = 0.01$ (see Appendix D.1.2 and Figure 5).

For comparison, the DKW inequality is applied to get a CDF lower bound, which is then transformed to an upper bound via the reduction approach in Section 4.1.1. To get the lower bound $b^l_{1:n}$, we set:

$$b^l_i = \max(0, \frac{\# \text{ points} \leqslant \frac{i}{n}}{n} - \sqrt{\frac{\log(\frac{2}{\delta})}{2n}})$$

### D.1.2 Numerical optimization examples (Section 5.1.2)

We parameterize the bounds with a fully connected network with 3 hidden layers of dimension 64. The $n$ gaussian seeds are of size 32, which is also the input dimension for the network. Training is performed in two stages, where the network is first trained to approximate a Berk-Jones bound, and then optimized for some specified objective $O$. In both stages of training we aim to push the training error to zero or as close as possible (i.e. "overfit"), since we are optimizing a bound and do not seek generalization. The model is first trained for 100,000 epochs to output the Berk-Jones bound using a mean-squared error loss. Then optimization on $O$ is performed for a maximum of 10,000 epochs, and validation is performed every 25 epochs, where we choose the best model according to the bound on $O$. Both stages of optimization use the Adam optimizer with a learning rate 0.00005, and for the second stage the constraint weight is set to $\lambda = 0.00005$. We perform post-processing to ensure the constraint holds (see Section A.2.2). For some denominator $m$ (in our case $m = 10^6$) we set $\gamma = \frac{1}{m}, \frac{2}{m}, \frac{3}{m}, ...$ and check the constraint until it is satisfied.

This approach is applied to both the experiments in Section 5.1.1 and Section 5.1.2. Details on the objective for Section 5.1.1 are above in Appendix D.1.1. In Section 5.1.2, we set $\delta = 0.01$ and our metrics for optimization are described below:

**CVaR** CVaR is a measure of the expected loss for the items at or above some quantile level $\beta$. We set $\beta = 0.75$, and thus we bound the expected loss for the worst-off 25% of the population.

**VaR-Interval** In the event that different stakeholders are interested in the VaR for different quantile levels $\beta$, we may want to select a bound based on some interval of the VaR $[\beta_{min}, \beta_{max}]$. We perform our experiment with $\beta_{min} = 0.5, \beta_{max} = 0.9$, which includes the median ($\beta = 0.5$) through the worst-case loss exluding a small batch of outliers ($\beta = 0.9$).

**Quantile-Weighted** We apply a weighting function to the quantile loss $\psi(p) = p$, such that the loss incurred by the worst-off members of a population are weighted more heavily.

**Smoothed Median** We study a more robust version of a median:

$$\psi(p; \beta) = \frac{1}{a\sqrt{\pi}} \exp(-\frac{(p - \beta)^2}{a^2})$$

with $\beta = 0.5$ and $a = 0.01$, similar to a normal distribution extremely concentrated around its mean. See Figure 5 for an illustration of such a weighting.

### D.2 Bounds on standard measures (Section 5.2)

This section contains additional details for the experiments in Section 5.2.

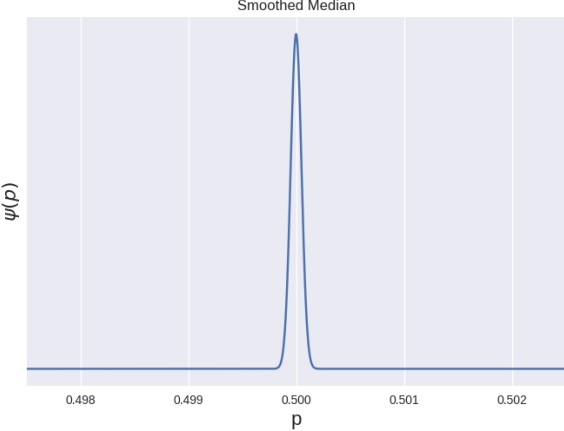

Figure 5: Plot of smoothed median function with $\beta = 0.5$ and $a = 0.01$

### D.2.1 RxRx1 (Section 5.2.1)

We use the code released by [17][8] to pre-train a model on the train split of RxRx1 [31] and we evaluate our algorithm on the OOD val split with 9854 total samples. We randomly sample 2500 items for use in validation (bounding and model selection), and use the remainder of the data points for illustrating the empirical distribution induced by the different hypotheses. The thresholds which are combined with the pre-trained model to form our hypothesis set are evenly spaced in $[-8, 0]$ under the log transformation with base 10, thus leaving the thresholds in the range $[10^{-8}, 1]$.

Balanced accuracy is calculated as:

$$L(\hat{Y}, Y) = 1 - \frac{1}{2}(\text{Sens}(\hat{Y}, Y) + \text{Spec}(\hat{Y}, Y)), \text{ where}$$

$$\text{Sens}(\hat{Y}, Y) = \frac{|\hat{Y} \cap Y|}{|Y|} \text{ and } \text{Spec}(\hat{Y}, Y) = \frac{k - |Y| - |\hat{Y} \setminus Y|}{k - |Y|}.$$

where $Y$ is the set of ground truth labels (which in this experiment will always be one label), $\hat{Y}$ is a set of predictions, and $k$ is the number of classes.

### D.2.2 MovieLens-1M (Section 5.2.2)

MovieLens-1M [12] is a publicly available dataset. We filter all ratings below 5 stars, a typical pre-processing step, and filter any users with less than 15 5-star ratings, leaving us with 4050 users. For each user, the 5 most recently watched items are added to the test set, while the remaining (earlier) items are added to the train set. We train a user/item embedding model using the popular python recommender library LightFM [9] with a WARP ranking loss for 30 epochs and an embedding dimension of 16.

For recommendation set $\hat{I}$ we compute a loss combining recall and precision against a user test set $I$ of size $k$:

$$L = \alpha l_r(\hat{I}, I)^2 + (1 - \alpha)l_p(\hat{I}, I)^2, \text{ where}$$

$$l_r(\hat{I}, I) = 1 - \frac{1}{k}\sum_{i \in I} \mathbb{1}\{i \in \hat{I}\} \text{ and } l_p(\hat{I}, I) = 1 - \frac{1}{|\hat{I}|}\sum_{i \in \hat{I}} \mathbb{1}\{i \in I\}$$

where $\alpha = 0.5$. We randomly sample 1500 users for validation, and use the remaining users to plot the empirical distributions. The 100 hypotheses tested are evenly spaced between the minimum and maximum scores of any user/item pair in the score matrix.

---

[8]https://github.com/p-lambda/wilds
[9]https://github.com/lyst/lightfm

# E    Additional results for numerical optimization (Section 5.1.2)

Figure 6 compares the learned bounds $G_{opt}$ to the Berk-Jones ($G_{BJ}$) and Truncated Berk-Jones ($G_{BJ-t}$) bounds, as well as the empirical CDF of the real loss distribution.

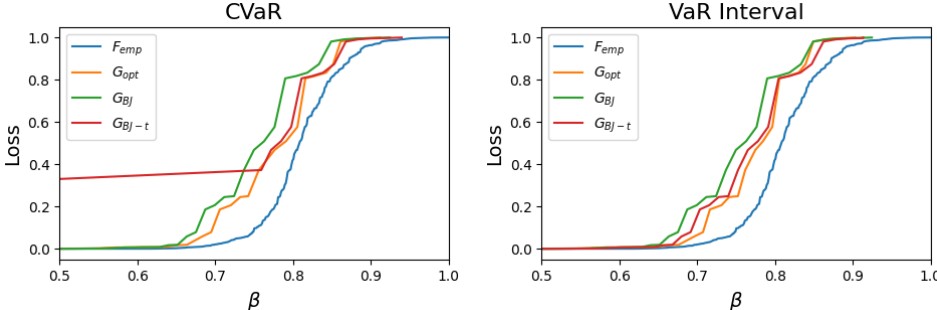

Figure 6: Learning tighter bounds on functionals of interest for protected groups. On the left, a bound is optimized for CVaR with $\beta = 0.75$, and on the right a bound is optimized for the VaR Interval $[0.5, 0.9]$. In both cases the optimized bounds are tightest on both the target metric as well as the mean, illustrating the power of adaptation both to particular quantile ranges as well as real loss distributions.

