# OpenReview forum: "Distribution-Free Statistical Dispersion Control for Societal Applications"
_NeurIPS.cc/2023/Conference — NeurIPS 2023 spotlight_

### Official Review · Reviewer_21Hg · 2023-07-05

**Soundness:** 3 good
**Presentation:** 3 good
**Contribution:** 3 good
**Rating:** 7
**Confidence:** 3

**Summary:**

This study extends the literature on distribution-free uncertainty quantification by providing bounds for various statistical dispersion measures, particularly those commonly employed in fairness evaluations of algorithms. The authors enhance the theoretical foundation by introducing additional results that allow bounding any functions with bounded total variation on the cumulative distribution function (CDF). Additionally, they propose a numerical optimization technique that achieve tighter bounds than previous methods. Empirical experiments validate the effectiveness of these tighter bounds and their relevance to ensuring fairness in model selection.

**Strengths:**

The paper addresses an existing gap in the literature on bounding other functions of the observed loss of an algorithm beyond means and QRBMs [1]. This is particularly relevant in the case of group-based measures that often appear in fairness analyses. The technical results that enable this are insightful and the proposed numerical optimization algorithm looks promising. The comprehensive empirical evaluation of the proposed methodology is commendable and provides valuable insights.

[1] Jake C Snell, Thomas P Zollo, Zhun Deng, Toniann Pitassi, and Richard Zemel. Quantile risk control: A flexible framework for bounding the probability of high-loss predictions. arXiv preprint arXiv:2212.13629, 2022.

**Weaknesses:**

1. Significance

	I am a bit torn on the significance of this contribution. The contribution is two-fold: provide the theoretical underpinnings for bounds of other types of functions on the CDF and provide a numerical optimization method that gives tighter bounds than the truncated Berk-Jones introduced in [1]. However, the technical results are not particularly insightful or novel and the bounds provided by the numerical optimization are comparable with the truncated Berk-Jones ones. On the other hand, the immediate relevance to fairness applications is noteworthy and this practical relevance alone makes the work valuable and suitable for presentation at the conference.

2. Paper structure

	The writing in its current form lacks sufficient focus and intuition. The authors include many results in the main paper, while relegating most of the technical and experimental insights to the appendix. Consequently, the paper becomes excessively dense in certain sections. To address this, I would suggest narrowing the focus to a select few measures and demonstrating how the novel results apply, while providing intuitive explanations. The remaining measures could be included in the appendix. Similarly, a more extensive discussion on the numerical optimization method would greatly enhance the paper's clarity. By adopting these suggestions, the paper can strike a better balance between comprehensive coverage and presenting key insights effectively.

**Questions:**

See weaknesses +:

* I understand that the numerical optimization gives tighter bounds, but how do you ensure that these bounds are valid?

**Limitations:**

Limitations were adequately addressed by the authors.

---

> ### Author Rebuttal · Authors · 2023-08-10
>
> Thank you for your thoughtful and overall positive feedback. Here, we provide answers to your concerns point by point.
>
> **C1:** I am a bit torn on the significance of this contribution...
>
> **A1:** We agree that our main focus is to extend existing methods to address societal applications more than technical novelty.
> We appreciate your recognition of our work's value with respect to algorithmic fairness.
>
> **C2:** The writing in its current form lacks sufficient focus and intuition...
>
> **A2:** We agree that our presentation of the material needs to be revised to better balance clarity and technical depth. Thanks for the suggestion about narrowing the focus and also discussing the numerical method more extensively. These will all be reflected in our revision.
>
> **C3:** I understand that numerical optimization gives tighter bounds, but how do you ensure that these bounds are valid?
>
> **A3:** Thanks for bringing up this point. We addressed this important question in the main paper but will highlight it in the revision. In lines 218-219 we have a post-processing procedure to ensure the bounds are valid. In Appendix A.2.2 between line 558-565, we provide details on how to post-process. This can be implemented simply via a binary search.

---

> > ### Comment · Reviewer_21Hg · 2023-08-22
> > **Response to Rebuttal**
> >
> > Thank you for addressing my concerns. I maintain that this work will be an important addition to the current body of work in this area. I have increased my score with the understanding that the authors will follow through with point A2 above in the final version of the paper.

---

### Official Review · Reviewer_Jihe · 2023-07-05

**Soundness:** 4 excellent
**Presentation:** 3 good
**Contribution:** 3 good
**Rating:** 6
**Confidence:** 4

**Summary:**

The paper demonstrates how to give distribution-free probabilistic control of certain statistical dispersion measures that are relevant in societal applications. An example is the GINI coefficient.

The paper technically builds on and slightly extends the work of Snell et al., which shows how to control quantile-based risk measures.

**Strengths:**

Congratulations to the authors on the great paper. It is the best in my batch of 6.
I am familiar with the line of work on quantile-based risk measures, and am happy to see it used in such an interesting application domain.
The GINI coefficient and friends are a unique set of application domains, with large practical importance. Though I do not necessarily see large methodological novelty from the statistical point of view in this paper, I do believe that it has the potential for large impact in the social sciences (politics/economics, etc.) if the techniques are adopted.


**Weaknesses:**

I will be an advocate for the paper's acceptance, but I would really encourage the authors to do a major rewrite during the revision stage. The writing and presentation needs work, in my opinion.
I have taken careful notes below. They are a bit stream-of-consciousness, so please forgive me for the tone.

WRITING FEEDBACK:

Section 4 reads really strangely to me right now.
It contains essentially no technical content, but is written in a technical way.
It’s more of an advertisement for the appendix. But it is vague, so I do not understand what has been proven, and what technical tools were used to do that until after I read the appendix (which 99% of your readers will NOT do).
E.g. Under “Control for absolute and polynomial functions” you say “we can control these” but don’t give a formal statement saying what that means. Also, is it a corollary of some main result, or is it a bespoke proof? We don’t know until we read the appendix.
For that reason, I’m not really learning anything from section 4 as a reader, which is not good.
If you need to make space, cut 4.3 which is not important for main text.
 There are many places where the math just doesn’t make sense, and false statements are made, which makes the paper confusing. (To be clear, I believe all the statements are formally true, but not always for the reason given.) Examples:
(1) “building blocks mentioned above as linear functionals of J: (i) nonlinear functions of J, i.e. \xi(J)”. What is this supposed to mean? J represents either F or F^- so \xi is a functional, not a function right? And you’re saying it’s nonlinear, so how can it be linear? As a side note, this “building blocks” section really didn’t work for me in terms of clarifying the later examples. It is so mathematical! Who are you writing for, a theorist, an applied statistician, or a social scientist? You have to decide, because the latter does not know what a function of a functional is, and the middle one doesn’t care and will probably be turned off by that language in my honest opinion.

As an alternative, just say that you are trying to control “functions of a whole CDF, like the GINI coefficient.” You can delineate the exact distinctions later when you’ve already made your main piont. “nonlinear function of a functional” and similar language is definitely distracting, not helping.

(2) Under “Control for a monotonic function”, this does not hold because of Proposition 1. Proposition 1 is a deterministic statement, and you need Proposition 3 to make the probabilistic statement. This is confusing because it is obviously false to say that “by proposition 1 you have X with probability 1-\delta”.

(3) The statement of Proposition 3 is false. The second line holds only with high probability.

(4) In Appendix C, it’s not “By classic DKW inequality” — it should say “By the classic DKW inequality and a Union bound/Bonferroni correction” because the classic DKW inequality does not have a k in it. Also, I find this subsection uninteresting and would consider greatly shortening it by just stating “you can use Bonferroni correction” and then giving the example.

The paper strikes a strange balance of being needlessly complex while presenting almost no theoretical detail in the main manuscript. It is fine to defer the most general form to the appendix, but in my honest opinion, the hybrid that is happening now is not very effective. Here are a few suggestions and places where I think the writing is ineffective.

(1) Simplify superscripts and subscripts. The letter “F” gets way too many superscripts/subscripts/hats. It is hard for an unfamiliar reader to keep track of what’s happening. But it can be simplified with essentially no loss: for example, because you never write F_{n/2} or F^{delta/2} at all in the main paper, it suffices to remove those scripts entirely.

(2) Remove technical minutiae. Remark 1 can be removed and the same point can be made as a half-sentence somewhere. Don’t use J as a generic notation for F or F^-, it’s very confusing.

(3) I would add at least one actual technical result in the main body of the paper. Everything is in the appendix right now. I definitely respect the reasoning behind that choice, but right now it really isn’t working. The reality is that the paper is very mathematical, but without some clear theoretical statement (Theorem 2 from appendix would suffice), it’s not clear what the main result is and why it works. Furthermore, things like Proposition 1 do not help. Proposition 1 is obvious (does not require proof) and it isn’t directly connected to any of the claims you make in the paper. It should definitely be in the appendix. Whereas Theorem 2 is your main hammer, so you should highlight it. And you should explicitly state somewhere that the

The sentence starting with “Roughly speaking, a function of bounded total variation…” is totally out of place given how technical the previous parts of the section have been.

**Questions:**

N/A, just let me know if I have misunderstood anything above.

**Limitations:**

Yes.

---

> ### Author Rebuttal · Authors · 2023-08-10
>
> We really appreciate your careful examination, thoughtful and overall positive feedback. We will address your comments below.
>
> **C1:** ...I would really encourage the authors to do a major rewrite during the revision stage. The writing and presentation need work...
>
> **A1:**  Thank you for your suggestions on presentation, in particular with respect to Section 4. We agree with your assessment of this section and plan a significant rewrite incorporating the suggestions.
>
> Meanwhile, for $\xi$, we realize that our formulation can be confusing. $\xi$ can be viewed as a function of $F$ or a function of $F(x)$ for some fixed $x$. If we view $\xi$ as a function of $F$ that induces $\xi(F(\cdot))$, then it is a function of a function, i.e. a functional. But $\xi$ is also a mapping from $R$ to $R$ once we fix an input $x$ and view $F(x)$ as a scalar. We will rewrite and clarify this point.
>
>
> We will also add $1-\delta$ in proposition 3 and simplify the subscripts and superscripts for estimators of $F$ in different cases. Again, thanks for your careful examination! As for Appendix C, we indeed should say ``by classic multi-dimensional DKW inequality in [2]". Appendix C meant to say we should take the best between multi-variate DKW and Frechet-Hoeffeding bounds and we will shorten that as suggested.
>
> Lastly, for a general answer, we agree with all the points brought up. Specifically, we agree that Section 4, for example, Section 4.1 is written in a hand-waving way because we put all the details in Appendix. We are also grateful for your suggestion about removing remark 1 and writing out the formal statement of Theorem 1 in a technical and formal way as suggested. These will all be reflected in our revision.
>
> [2] Naaman et. al. "On the tight constant in the multivariate Dvoretzky–Kiefer–Wolfowitz inequality"

---

> > ### Comment · Reviewer_Jihe · 2023-08-10
> > **Great job again**
> >
> > Looks like the paper is in good shape. I've read the response and get the sense that the authors will make a significant revision to the writing, which was my main concern.
> > I see no reason the paper should not be accepted at this point. (But please, do follow through on the feedback.)
> > Congratulations again.

---

> > > ### Author Response · Authors · 2023-08-16
> > > **Further response to reviewer Jihe**
> > >
> > > Thank you so much for your kind words and helpful suggestions!

---

### Official Review · Reviewer_6JyR · 2023-07-25

**Soundness:** 4 excellent
**Presentation:** 3 good
**Contribution:** 3 good
**Rating:** 7
**Confidence:** 3

**Summary:**

Given a trained predictor, this paper proposes a framework for bounding various measures of dispersion of the loss distribution.

The approach taken here is similar to that of conformal prediction, where one uses a calibration set to estimate the distribution of the loss. In this paper, the authors use the empirical CDF of the loss function on the calibration set to build confidence bands for the true (unknown) CDF. These confidence bands can then be used to obtain bounds on the dispersion measure of interest.  This approach is demonstrated on several measures of dispersion: Gini coefficient, Atkinson index, difference of group medians, and others.


**Strengths:**

* This paper addresses an important class of problems.
* The proposed methodology has a broad scope as it is applicable to any ML system and any data distribution (so long as it is i.i.d. and no distribution shift has occurred).
* The method should be easy to incorporate into standard ML packages and workflows. Thus, it has a high potential impact on ML practice.
* In terms of originality, this is a moderate contribution and seems mostly like a follow-up paper to [1]. However its applicative focus is sufficiently different. Also, it contains some technical novelty. Chiefly, the gradient-based optimization approach for the construction of tight confidence bands for a given dispersion measure.
* On the experimental front, the proposed approach is demonstrated on a diverse set of dispersion measures and data sets, showing its real-world applicability.

[1] Jake Snell, Thomas P Zollo, Zhun Deng, Toniann Pitassi, Richard Zemel. "Quantile Risk Control: A Flexible Framework for Bounding the Probability of High-Loss Predictions" ICLR (2023).



**Weaknesses:**

The results in this paper are based on the construction of confidence bands for the CDF of the loss and in particular on the use of goodness-of-fit statistics for the construction of these bands. This idea is well-known in the field of statistics and the authors should do a better job of connecting with that literature. In particular, the paper [2] from 1995 proposed constructing confidence bands for the CDF using the Berk-Jones statistic, just like the baseline used in the current paper, but is not cited.

Minor points for improvement:
1. The fonts used in the figures are a bit small.
2. The notation in Section 2 is confusing. I believe it would be better to use the standard notation: X for inputs and Y for outputs. Z can be used for loss(h(X),Y).
3. The last paragraph of Section 3 is a bit much. Consider breaking it down and providing specific simple examples for clarity.
4. Page 4, line 139: a space is missing in "measures.They"

[2] Art Owen. "Nonparametric likelihood confidence bands for a distribution function". Journal of the American Statistical Association (1995)

**Questions:**

* In many applications, one is given a finite dataset to work with. In which case, under the current paper's framework there is a need to decide how to split the data into a primary training set and an additional calibration set (which the authors refer to as validation set, however this term is different from that used in relation to cross-validation). Are there any guidelines that you can provide on this matter? e.g. minimal size of the calibration needed set to achieve a certain error.

* Using a fully-connected NN to parameterize the confidence boundaries in the optimization seems like overkill. I suggest that the authors try a much simpler parameterization, such as low-degree polynomials or using the first few elements of a Fourier series. This should be sufficient to express any smooth boundary accurately and is likely to converge faster due to the smaller number of parameters.

**Limitations:**

* This paper only considers one-sided confidence bands for the CDF and the bounds on dispersion that result from them. In some cases it may be sensible to consider two-sided confidence bands but this point is not discussed in the paper. It seems that the repeated integration method for one-sided bands on which this paper is based can only be used for computing one-sided bands. I think this limitation should be at least mentioned.  One could approximate a two-sided 1-alpha level boundary by combining an upper and lower band at the level of 1-alpha/2. However, this is not optimal (and is also not mentioned in the paper). Note that there are methods in the literature that compute two-sided confidence bands for the CDF that are based on similar ideas and they could be incorporated in some manner into the framework presented in the paper (but this is too much to ask for in a revision!).

* Distribution shift is not mentioned anywhere, though I don't see this as a clear limitation of the proposed framework. Depending on the type of distributional change, one might be able to construct wider confidence bands that take it into account.

---

> ### Author Rebuttal · Authors · 2023-08-10
>
> Thank you for your constructive suggestions and overall positive feedback. We will revise our paper according to your helpful suggestions and will include the important citations we missed. Here, we provide answers to your other concerns point by point.
>
> **C1:** In many applications... under the current paper's framework there is a need to decide how to split the data into a primary training set and an additional calibration set (which the authors refer to as validation set... are there any guidelines that you can provide on this matter? e.g. minimal size of the calibration needed set to achieve a certain error.
>
> **A1:** While data splitting was an important issue in some older work in Conformal Prediction, we adopt the setting used in most recent work in distribution-free uncertainty quantification (DFUQ) [1,2].  In this setting, we start with a pre-trained blackbox model, and only need to use a validation dataset to calibrate and choose a hypothesis (for example, a threshold), without considering the training process for the blackbox model.
>
> [1] Snell et. al. "Quantile Risk Control: A Flexible Framework for Bounding the Probability of High-Loss Predictions."
>
> [2] Angelopoulos et.al. ``Learn then Test: Calibrating Predictive Algorithms to Achieve Risk Control."
>
> **C2:** Using a fully-connected NN to parameterize the confidence boundaries in the optimization seems like overkill.
>
> **A2:** Thank you for the suggestion.  We implemented a polynomial model with different levels of complexity for comparison.  For $L_1,L_2,..., L_n$, we sample $n$ one-dimensional Gaussian seeds denoted as $s_1,s_2,...,s_n$. Then:
> $\phi_{\theta}(s_i)=\theta_0+\theta_1 s_i + \theta_2 s_i^2 + ... + \theta_k s_i^k.$
>
> Results with the polynomial model after some hyperparameter tuning are reported in Table 1 in the PDF attached in the aruthor rebuttal.  Overall we see mixed performance from the polynomial parameterization.  It produces the best bound on the smoothed-median metric, and performs better than Berk-Jones on CVaR.  However, it fails to produce a better bound than Berk-Jones for the other two metrics.
>
> Overall, we find the neural network to be easy to implement and optimize -- importantly, the same set of parameters are used for learning bounds on all target metrics in Sections 5.1.1 and 5.1.2.  Further, the networks can be trained in a few minutes on a single small GPU. While other parameterizations are possible, and sufficient hyperparameter tuning may lead to better performance across the board for the polynomial method, we believe the neural network is a good choice based on the robust results it produced in our experiments and the reasonable compute and memory costs.
>
> **C3:** This paper only considers one-sided confidence bands for the CDF and the bounds on dispersion that result from them...
>
> **A3:** Our framework indeed can provide two-sided bounds. This is illustrated in
> Figure 1, where we plotted two-sided bounds for Lorenz curve. Lemma 1 converts the lower bound of CDF obtained by [1] to an upper bound; both bounds hold simultaneously without incurring any inflation factor. We will clarify this in the revision.
>
> **C4:** Distribution shift is not mentioned anywhere, though I don't see this as a clear limitation of the proposed framework...
>
> **A4:** We did mention the assumption of consistency of the validation and test distributions as a limitation of the work. We agree that this is an important consideration.

---

> > ### Comment · Reviewer_6JyR · 2023-08-11
> >
> > Overall I am happy with the rebuttal. Considering all the other reviews and your stated intentions of revising the text I think the resulting paper would be very nice.
> >
> > I have one comment and one question:
> >
> > (C2) It is great that you tried running a simple parametric representation. I think it would be helpful to have a plot in the revised Appendix that compares the bounds obtained using polynomials to the other approaches.
> >
> > (C3) Could you please clarify this here? If you construct a 1-alpha two-sided band by taking a 1-alpha/2 lower band and a 1-alpha/2 upper band then the two-sided band you obtain would be conservative (i.e. it would hold with probability greater than 1-alpha).

---

> > > ### Author Response · Authors · 2023-08-16
> > > **Additional response to reviewer 6JyR**
> > >
> > > Thanks for your feedback.
> > >
> > > For (C2), yes, we will include that in our revision.
> > >
> > > For (C3), thanks for the question. First,  it is true that typical two-sided bounds statements in literature are with probability $\ge 1-\alpha$, which is standard in applying concentration inequalities though might be conservative. But we guess you are asking a slightly different question and here is our tentative answer to your possible concern. Please let us know if it doesn't address your concerns and we would love to further try our best to clarify.
> > >
> > > We guess what you are concerned about is the following problem: if one only considers one-sided bound, **for example**, $1-\alpha$ one-sided DKW inequality for CDF $F$, then $\hat F^\alpha_{n, L}=\hat F - C\sqrt{\frac{\ln(1/\alpha)}{n}}$. However, if we consider two-sided bounds with $1-\alpha/2$ for the upper band and lower band, then
> > >
> > > $F^{\alpha/2}_{n,L} =\hat F - C\sqrt{\frac{\ln(2/\alpha)}{n}}$ ,
> > >
> > > $F^{\alpha/2}_{n,U}=\hat F +C\sqrt{\frac{\ln(2/\alpha)}{n}}$.
> > >
> > >
> > > From here, we can see the band obtained by  $\hat F^{\alpha/2} _ {n,L}$    is wider than    $\hat F^\alpha _ {n,L}$.
> > >
> > > This above problem can be addressed by our Lemma 1 because we consider constructing bounds via $L_1,...L_n$. Specifically, our Lemma 1 states that any $1-\alpha$ lower bound constructed by $L_1,...L_n$ for $F$ (upper bound for $F^-$) can be transformed to an upper bound for $F$, and the upper bound will automatically hold as long as the lower bound holds. That means with probability at least $1-\alpha$, both upper and lower bounds hold and the lower bound will not be worse compared to the construction in [1] for one-sided lower bound.

---

### Official Review · Reviewer_DzT2 · 2023-07-26

**Soundness:** 4 excellent
**Presentation:** 3 good
**Contribution:** 3 good
**Rating:** 7
**Confidence:** 5

**Summary:**

The paper proposes a methodology for distribution-free confidence intervals for a wide class of dispersion measures. This is motivated by validating the performance of machine learning algorithms with respect to more complex notions of performance, such as group-based measures of loss balance or the Gini coefficient of the allocation of the losses. This paper studies the problem in more generality than previous work, which focused on losses that are empirical means of quantities or quantile-based risk measures. The paper demonstrates the proposed methodology on several solid benchmark tasks.

**Strengths:**

The problem is well-motivated, and the need for such guarantees about (dispersion) risk measures is convincing. Such techniques would help audit and verify aspects of the performance of machine learning algorithms. I find the introduction and abstract to be clear and set the scene well.

The work involves some non-trivial mathematics, and it is a formidable technical extension of previous work. A reader of previous work would not be able to apply it to the risk measures studied herein without this manuscript.

The examples are serviceable and demonstrate the breadth of the proposed method.



**Weaknesses:**

My main concern is with the technical presentation in Section 4. I am fairly knowledgeable about this literature, but I still found it challenging to read at first. The authors have a lot of technical content to cover with limited space, so right now much of it is deferred to the appendix. I understand that this is challenging, and I appreciate all the work that has gone into this draft. Nonetheless, I hope this can be improved in the next iteration. In general, my opinion is probably too much content for the amount of space that is currently allocated, so perhaps more space can be made for this section.

My first suggestion is that the authors state the desired property of the output of their algorithm explicitly at the beginning of Chapter 4 or perhaps earlier -- if I understand correctly the goal is to give a 1-delta confidence interval for some risk measure. The output of the algorithm is this confidence interval. This is in contrast to some other works in risk control and conformal prediction, where the output of the algorithm is a choice of a parameter that controls the coverage rate or other risk level. When the authors of this work use the word "control" in the abstract and at various other points, I was initially confused as to what they meant.

Secondly, it's not immediately clear what the proposed algorithm is when reading section 4. The section seems quite compressed, and as a result the reader is left with a lot of work to put together the pieces from the relatively lean presentation. I would find it helpful to see an algorithm environment or pseudocode explaining exactly the steps to compute the upper bound for at least one of the cases considered.

It goes without saying that when it comes to presentation choices, it is somewhat a matter of taste. I understand the authors may wish to go a different direction than my suggestions above. Still, iterating on this initial draft would be very helpful. In good faith, I'm giving the paper a score of "7" in anticipation that the final version incorporates some improvements to the presentation.

**Questions:**

Theorem 1 seems like the technical core. Is this impression correct? If so, more commentary about the functions f_1 and f_2 would be helpful. These exist in principal, but for which cases are they actually tractable?

There is currently some notational pain with the overloading of \xi. Sometimes it is a function R to R, sometimes it is a functional. 4.1 is titled "Control of nonlinear functions ..." but then starts off with "xi(F^-) which maps F^- to another function of R", which is interpretting it as a function. However, later on it is said that xi is monotonic, which is invoking the interpretation of xi as a function R to R rather than a function. Similarly, 4.2 is about "control of nonlinear functionals" but then in the integral expression in line 183, F^- is evaluated at alpha, which means this expression is uses \xi as a function, not functional.  I don't doubt that it's all correct, but it was hard for me to read at first. It would help the object \xi is defined explicitly somewhere, so that exactly what type of object (function vs functional) it is is stated.

Minor:
-- Berk-Jones and DKW should be cited.
-- I know of two other references in the conformal prediction literature that use uniform bounds on the CDF like Berk-Jones; https://arxiv.org/abs/2104.08279 and https://arxiv.org/abs/2304.06158 . These works are somewhat technically related, although I understand the use of the bounds on the loss distribution is conceptually different than using the bounds on the conformal score function.



**Limitations:**

The authors do a good job explaining the context and limitations of their work. I would say the main limitation is that these techniques (like most techniques in this field) do not handle distribution shifts, and are assuming an iid data sample. This is explicitly addressed in the conclusion.

---

> ### Author Rebuttal · Authors · 2023-08-10
>
> Thank you for your thoughtful and overall positive feedback. Here, we provide answers to your concerns and questions point by point.
>
> **C1:**...main concern is with the technical presentation in Section 4...have a lot of technical content to cover with limited space, so right now much of it is deferred to the appendix...
>
> **A1:** Thank you, we agree with your assessment of this section and plan a significant revision.
>
> **C2:** My first suggestion is that the authors state the desired property of the output of their algorithm explicitly at the beginning of Chapter 4...if I understand correctly the goal is to give a 1-delta confidence interval for some risk measure...This is in contrast to some other works in risk control and conformal prediction, where the output of the algorithm is a choice of a parameter that controls the coverage rate or other risk level....
>
> **A2:** We use the term control in the same way as these other works -- our framework can be applied to choose a hypothesis that best controls some risk measure. We will clarify this in the revision. We would like to point out a distinction between how we exercise this control versus other recent approaches, notably ``learn then test" [1]. In both, as you state, the most important step is to provide a ($1-\delta$)-confidence upper bound for some risk measure for a given hypothesis $h$.
> In [1] they first fix a risk level $\alpha$ and output all hypotheses whose risk (their risk is defined as the mean of some loss function) is upper bounded by $\alpha$.
> Our algorithm iterates over all members in $\mathcal H$ and selects a hypothesis $h$ corresponding to the smallest risk upper bound. But we our framework can naturally be used in their setting since we also obtain bounds for risks and those bounds hold simultaneously for all hypotheses with high probability.
>
> [1] Angelopoulos et.al. ``Learn then Test: Calibrating Predictive Algorithms to Achieve Risk Control."
>
> **C3:** ...it's not immediately clear what the proposed algorithm is when reading section 4. The section seems quite compressed, and as a result the reader is left with a lot of work to put together the pieces from the relatively lean presentation. I would find it helpful to see an algorithm environment or pseudocode...
>
> **A3:** Thanks for the great suggestion to summarize our algorithm in pseudo-code -- we will include this in our revision. Our algorithm can utilize any method that produces high probability two-sided bounds for the underlying CDF, where Berk-Jones and our numerical optimization bounds are two examples. The pseudo-code will make this clear.
>
> **C4:** Theorem 1 seems like the technical core. Is this impression correct? If so, more commentary about the functions $f_1$ and $f_2$ would be helpful. These exist in principal, but for which cases are they actually tractable?
>
> **A4:** You are right that Theorem 1 is one of our main results. This result can extend existing results in the literature to bound the loss beyond means and quantile-based risks.
> We have a formal version of Theorem 1 in Appendix line 513-517.
> $f_1$ and $f_2$ are quite tractable, under mild assumptions. For example, if $\xi$ is continuously differentiable, $f_1(x)=\int^x_0|\frac{d\xi}{ds}(s)|ds$ and $f_2(x)=f_1(x)-\xi(x)$.
>
> **C5:** There is currently some notational pain with the overloading of $\xi$......Berk-Jones and DKW should be cited...
>
> **A5:** Thanks for your other suggestions for revisions and notation clarifications. And for the pointers to missing citations. We will include these in our revision.

---

### Author Rebuttal · Authors · 2023-08-10

We thank all four reviewers for their constructive and positive feedback, and for thinking our paper addresses important applications, has the potential for significant impact on social science, and addresses existing gaps in the literature on bounding the observed loss beyond means and quantile-based risks.

We are very grateful for the extensive useful suggestions on how to improve the presentation, especially for Section 4.  Although we cannot upload a revised draft during the rebuttal phase due to the NeurIPS policy, we plan to modify our paper accordingly. We also address concerns (**C**) and provide our answers (**A**) for each reviewer individually below.

In addition, we attach a PDF for extra experimental results.

---

### Decision · Program_Chairs · 2023-09-21

**Decision:**

Accept (spotlight)

**Comment:**

All reviewers agree that the paper makes a substantial and interesting contribution to the area of construction distribution free confidence intervals for a class of practically important problems.